# Enhanced Trans-Himalaya Pollution Transport to the Tibetan Plateau by the Cut-off Low System

Ruixiong Zhang[1], Yuhang Wang[1], Qiusheng He[2], Laiguo Chen[3], Yuzhong Zhang[1], Hang Qu[1], Charles Smeltzer[1], Jianfeng Li[1], Leonardo M.A. Alvarado[4], Mihalis Vrekoussis[4,5,6], Andreas Richter[4], Folkard Wittrock[4], and John P. Burrows[4]

[1]School of Earth and Atmospheric Sciences, Georgia Institute of Technology, Atlanta, GA
[2]School of Environment and Safety, Taiyuan University of Science and Technology, Taiyuan, China
[3]Urban Environment and Ecology Research Center, South China Institute of Environmental Sciences (SCIES), Ministry of Environmental Protection (MEP), Guangzhou, China
[4] Institute of Environmental Physics and Remote Sensing, University of Bremen, Bremen, Germany
[5] Center of Marine Environmental Sciences – MARUM, University of Bremen, Germany
[6] Energy, Environment and Water Research Center (EEWRC), The Cyprus Institute, Nicosia, Cyprus

*Correspondence to*: Yuhang Wang (ywang@eas.gatech.edu), Qiusheng He (heqs@tyust.edu.cn)

**Abstract.** Long-range transport followed by deposition of black carbon on glaciers of Tibet is one of the key issues of climate research inducing changes on radiative forcing and subsequently impacting the melting of glaciers. The transport mechanism, however, is not well understood. In this study, we use short-lived reactive aromatics as proxies to diagnose transport of pollutants to Tibet. In situ observations of short-lived reactive aromatics across the Tibetan Plateau are analyzed using a regional chemistry and transport model. The model performance using the current emission inventories over the region is poor due to problems in the inventories and model transport. Top-down emissions constrained by satellite observations of glyoxal are a factor of 2-6 higher than the a priori emissions over the industrialized Indo-Gangetic Plain. Using the top-down emissions, agreement between model simulations and surface observations of aromatics improves. We find enhancements of reactive aromatics over Tibet by a factor of 6 on average due to rapid transport from India and nearby regions during the presence of a high-altitude cut-off low system. Our results suggest that the cut-off low system is a major pathway for long-range transport of pollutants such as black carbon. The modeling analysis reveals that even the state-of-the-science high-resolution reanalysis cannot simulate this cut-off low system accurately, which probably explains in part the underestimation of black carbon deposition over Tibet in previous modeling studies. Another model deficiency of underestimating pollution transport from the south is due to the complexity of terrain, leading to enhanced transport. It is therefore challenging for coarse-resolution global climate models to properly represent the effects of long-range transport of pollutants on the Tibetan environment and the subsequent consequence for regional climate forcing.

# 1 Introduction

The Tibetan Plateau, commonly referred as the Third Pole and the last pristine land of the Earth, has drawn much attention in environmental and climate research in recent years (Menon et al., 2002). Although Tibet appears to be isolated from industrialized regions due in part to the transport barrier by its being a plateau and its pollutant concentrations being generally low, the Third Pole is vulnerable to regional climate change. Areas of the Tibetan Plateau over 4 km in altitude are warming at a rate of 0.3 °C per decade, twice as fast as the global average (Xu et al., 2009). In addition to the increase of greenhouse gases (GHGs) and the associated global warming, black carbon (BC) is likely another important contributor to the warming of the Tibetan Plateau. The deposition of BC on the vast glaciers of the Tibetan Plateau will decrease the surface albedo, accompanied by increased sunlight absorption and subsequent enhanced melting (Hansen and Nazarenko, 2004; Ramanathan and Carmichael, 2008; Ming et al., 2009; Yasunari et al., 2010). Increasing BC concentrations were previously found in ice core and lake sediment records (Xu et al., 2009; Cong et al., 2013). The dwindling of glaciers over Tibet is a major concern for fresh water supply to a large portion of the Asian population through the Indus River, Ganges River, Yarlung Tsangpo River, Yangtze River and Yellow River (Singh and Bengtsson, 2004; Barnett et al., 2005; Lutz et al., 2014). Though melting glaciers favor river runoff temporarily, mass loss of glaciers endangers water supply during the dry season in the future (Yao et al., 2004; Kehrwald et al., 2008).

Besides narrowing the uncertainties of BC emissions, aging and deposition, better understanding the transport pathways are equally important in this region. Surrounded by the largest BC sources of East Asia and South Asia (Bond et al., 2007; Ohara et al., 2007), Tibet is primarily affected by pollutant transport from these two regions (Kopacz et al., 2011; Lu et al., 2012; Zhao et al., 2013; Wang et al., 2015; Zhang et al., 2015; Li et al., 2016; Wang et al., 2016; Kang et al., 2016). Kopacz et al. (2011) attempted to identify the sources of BC over glaciers in the Himalayas and the Tibetan Plateau (HTP) using the adjoint model of GEOS-Chem. Lu et al. (2012) developed a novel back-trajectory model with BC emissions, hydrophilic-to-hydrophobic aging, and deposition and found that South Asia and East Asia account for 67% and 17% of BC over the HTP. Using source tagging, biofuel and biomass burning emissions from South Asia are found to be the largest sources of BC in HTP followed by fossil fuel combustion emissions (Zhang et al., 2015). Hindman and Upadhyay (2002) suggested that the vertical lifting due to convection and subsequent horizontal mountain-valley wind lead to the transport of aerosols from Nepal to Tibet. Dumka et al. (2010) also stressed the importance role of mountain-valley wind in BC concentration in Central Himalayas. Cong et al. (2015) suggested that both the large-scale westerlies from South Asia and the local mountain-valley wind from South Asia are major transport pathways. The synoptic scale trough and ridge can potentially lead to the trespassing of atmospheric brown clouds from South Asia to the Tibetan Plateau (Lüthi et al., 2015). Ji et al. (2015) indicated that the southwesterlies during monsoon season favor aerosols transport across the Himalayas from South Asia. Aerosols observations in previous studies are mostly limited to the southern and northern slopes of the Himalayas (Hindman and Upadhyay, 2002; Dumka et al., 2010; Cong et al., 2015) with very few in situ sites (e.g. Namco, Linzhi) inside Tibet (Kopacz et al., 2011; Ji et al., 2015; Lüthi et al., 2015; Zhang et al., 2015). Considering the complex topography (Lawrence and Lelieveld, 2010;

Ménégoz et al., 2013; He et al., 2014; Kumar et al., 2015) and scarce observations (Maussion et al., 2011), it is crucial to evaluate model simulated transport performance over the Tibetan Plateau using available observations with a good spatial coverage. Observation-constrained modeling is needed to better understand potential model biases due to the uncertainties of model simulated transport from South Asia to Tibet.

In this study, we use short-lived reactive aromatics as proxies to diagnose transport of pollutants to Tibet. In situ observations of short-lived reactive aromatics across the Tibetan Plateau are analyzed (Section 2.1). Anthropogenic emissions including fossil fuel combustion, gasoline evaporation and solvent use constitute the main sources of atmospheric aromatics (Sack et al., 1992; Fu et al., 2008; Henze et al., 2008; Cagliari et al., 2010; Cabrera-Perez et al., 2016). Biofuel and biomass burning is another important source (Fu et al., 2008; Henze et al., 2008). The main sink of aromatics is OH oxidation with lifetimes ranging from hours to days (Atkinson, 2000; Liu et al., 2012b). We use satellite observations to minimize the biases of emission inventories for upwind regions of Tibet (Sections 2.2, 2.3 and 2.4) and then apply a regional chemistry and transport model constrained by high-resolution reanalysis meteorological data to understand missing transport processes in model simulations (Section 3). On the basis of these results, we examine the implications for global climate modeling studies of anthropogenically driven changes over the Tibetan Plateau (Section 4).

## 2 Methods

### 2.1 In situ aromatics data

Whole air samples were collected in 2-L electro-polished stainless-steel canisters, which had been cleaned and vacuumed according to the TO-15 method issued by US EPA before shipment to the sampling sites. The restricted grab sampler (39-RS-x; Entech), which has a 5-μm Silonite-coated metal particulate filter, was placed on the inlet of the canister to completely filter out dust and other particulates during sampling. These samples were taken in daytime from 8:00 AM to 7:00 PM with an interval of 1 to 2 hours. The sampling time was 5 minutes to fill the vacuumed canisters. The filled canisters were transported back to the laboratory of Guangzhou Institute of Geochemistry, Chinese Academy of Science. Each air sample was analyzed for 65 light non-methane hydrocarbons (NMHCs) species. The samples were pretreated by an Entech Model 7100 Preconcentrator (Entech Instruments Inc., California, USA), and analyzed by a gas chromatography-mass selective detector (GC-MSD/FID, Agilent 7890A/5973N, USA) using dual columns and dual detectors to simultaneously analyze both low- and high-boiling-point VOCs with each injection. The detailed analytical procedure is described by Zhang et al. (2012).

In this study, we analyze 65 measurements of aromatics (benzene, toluene, ethyl-benzene, m/p/o-xylene) and wind speed measurements taken across Tibet during October 2010 (Fig. 1a). Care was taken in sampling such that there are no direct urban, industrial, or road emissions in the upwind direction of the sampling location. The lifetimes of toluene, ethyl-benzene, m/p/o-xylene are relatively short (2-20 hours) and these reactive aromatic compounds therefore provide observational constraints for transport from India and nearby regions to Tibet. We group the samples into three periods based on the time and locations of the measurements, i.e. Period 1 from October 13 to 17, 2010 to the north of the Himalayas along the southern

border of Tibet, Period 2 from October 19 to 24 across the interior of the Tibetan Plateau, and Period 3 of October 25 to the west of the Yarlung Tsangpo Grand Canyon in southeastern Tibet (Fig. 1a).

## 2.2 SCIAMACHY CHOCHO measurements

The SCanning Imaging Absorption spectroMeter for Atmospheric CHartographY (SCIAMACHY) onboard Environmental Satellite (ENVISAT) operated from 2002 to 2012 (Burrows et al., 1995, Bovensmann et al., 1999), with an overpass time at about 10:00 AM local time. SCIAMACHY made passive remote sensing measurements of the upwelling radiation from the top of the atmosphere in alternate nadir and limb viewing geometry. Mathematical inversion of the measurements of SCIAMACHY yields a variety of data products including glyoxal (CHOCHO) Vertical Column Densities (VCDs, unit: $molecules\ cm^{-2}$). The retrieval uses the Differential Optical Absorption Spectroscopy (DOAS) technique (Wittrock et al., 2006; Vrekoussis et al., 2009; Alvarado et al., 2014). The CHOCHO retrieval used in this study is based on the algorithm developed in Alvarado et al. (2014), which includes corrections for the interferences with nitrogen dioxide ($NO_2$) over the regions with high NOx emissions as well as liquid water over oceans (Alvarado, 2016). Detection limit for SCIAMACHY CHOCHO VCD is about $1 \times 10^{14}\ molecules\ cm^{-2}$. The overall monthly uncertainty of CHOCHO VCDs ($C_{CHOCHO}^{SCIAMACHY}$) in the selected region during October 2010 is given by $\alpha \times C_{CHOCHO}^{SCIAMACHY} + 1 \times 10^{14}\ molecules\ cm^{-2}$, where the value of $\alpha$ is in a range of 0.1 to 0.3. Following the method as described by Liu et al. (2012b), we derive a top-down aromatics emission estimate for South Asia constrained by CHOCHO retrievals described in Section 2.4.

## 2.3 3-D REAM model

We use the 3-D Regional chEmical trAnsport Model (REAM) to examine the chemistry evolution and regional transport of aromatics. REAM was used in previous studies, including large-scale transport (Wang et al., 2006; Zhao et al., 2009b, 2010), vertical transport (Zhao et al., 2009a; Zhang et al., 2014, 2016), emission estimates (Zhao and Wang, 2009; Liu et al., 2012b; Gu et al., 2013, 2014, 2016) and other air quality studies (Zeng et al., 2003, 2006; Choi et al., 2005, 2008a, 2008b; Wang et al., 2007; Liu et al.,2010, 2012a, 2014; Gray et al., 2011; Yang et al., 2011; Zhang and Wang, 2016).

REAM has a horizontal resolution of 36 km with 30 vertical levels in the troposphere and 5 vertical levels in the stratosphere covering adjacent regions of China (Fig. 1b). The model top is at 10 hpa. Meteorological fields in REAM are obtained from the Weather Research and Forecasting model (WRF) assimilations constrained by National Centers for Environmental Prediction Climate Forecast System Reanalysis (NCEP CFSR, Saha et al., 2010) 6-hourly products, which have a horizontal resolution of T382 (~38 km). We run the WRF model with the same resolution as in REAM with a domain larger than that of REAM by 10 grid cells on each side. Meteorological inputs related to convective transport are updated every 5 minutes while the others are updated every 30 minutes. The recent update of REAM expands the GEOS-Chem standard chemical mechanism (V9-02) to include a detailed description of aromatics chemistry (Bey et al., 2001; Liu et al., 2010, 2012b). Aromatics are lumped into three species based on reactivity, i.e. ARO1 (toluene, ethyl-benzene), ARO2 (m/p/o-xylene), and benzene. The atmospheric lifetimes of the three aromatics tracers against OH are 18 hours, 4.2 hours and 3.9 days during the study period

(October 13-25, 2010), respectively. Due to the long atmospheric lifetime of benzene, it is more difficult to track and identify its sources; thus we do not explicitly discuss benzene in this study. We focus our analysis on reactive aromatics (toluene, ethyl-benzene, and m/p/o-xylene).

Initial and boundary conditions for chemical tracers are taken from GEOS-Chem (V9-02) $2° \times 2.5°$ simulation (Bey et al.,

2001). Anthropogenic emissions are from the MIX inventory for October 2010 (Li et al., 2015). MIX is a mosaic Asian anthropogenic emission inventory with the Multi-resolution Emission Inventory for China (MEIC) for China and several other emission inventories for other Asian countries. In addition to the MIX inventory, we also conduct sensitivity simulations using the Intercontinental Chemical Transport Experiment-Phase B (INTEX-B) emissions inventory (Zhang et al., 2009; Li et al., 2014), which was developed for the year 2006. We find that compared to the in situ observations of aromatics, the simulation

results using the INTEX-B emissions are better. The main reason for the simulation improvements is due to the emissions of aromatics in South Asia. Given the large uncertainties in the emissions of aromatics (e.g., Liu et al., 2012b), this result is not surprising. Since MEIC and INTEX-B inventories are developed by the same group, we replace MIX aromatics emissions outside China with INTEX-B data such that aromatics emissions in the model are consistent. The improvements of model simulations compared to in situ observations are shown in Fig. S1 in the Supplement. Since satellite observations are used to

improve aromatics emissions (next section), using either MIX or INTEX-B emissions in this work gives the same conclusions. Biogenic VOC emissions are computed with the Model of Emissions of Gases and Aerosols from Nature (MEGAN) algorithm (v2.1, Guenther et al., 2012) and outdoor biomass burning emissions of CHOCHO and other species are based on Global Fire Emissions Database Version 4.1 with small fires (GFED4.1s, van der Werf et al., 2010; Andreae and Merlet, 2001; Lerot et al., 2010). Indoor burning CHOCHO emissions of India are computed using emission factors from Pettersson et al. (2011) and

Li et al. (2014). Rural and urban population distributions of India for year 2010 are used as spatial proxies (Balk et al., 2006; CIESIN, 2011, 2016). We adopt the energy consumptions for rural and urban inhabitants on the basis of the National Sample Survey Office of India (N.S.S.O., 2012a, 2012b). Compared with satellite observed CO and $NO_2$ VCDs, REAM performs reasonably well in the study region during October 2010 (Fig. S2 in the Supplement). For general model evaluations of REAM, we refer the readers to the papers cited early in this section.

We updated the INTEX-B emission inventory in South Asian countries through inverse modeling constrained by SCIAMACHY CHOCHO VCDs (next section). We run REAM simulations with the a priori and top-down emission inventories, and compare the results with observations in Section 3.1 and 3.2, respectively. We find that some of the model low bias is likely due to emission underestimation. We further carried out three model sensitivity tests to calculate the contributions to surface aromatics from emissions over Tibet, other provinces of China, and South Asia (India and nearby

regions). Each simulation is run with only the aromatics emissions from the corresponding region. The OH concentrations in each simulation are specified to the archived values of the full model simulation. The results for two sub-periods of Period 2 are examined in Section 3.3.

## 2.4 Top-down aromatics emission estimation

Compared with SCIAMACHY data, REAM using the original emission inventories archived at the overpass time of SCIAMACHY underestimates CHOCHO VCDs in the populated regions of India (Fig. 2). This underestimation is especially significant in the Indo-Gangetic Plain located south of the Himalayas (Fig. 2c). We then derive the top-down aromatics emissions for these regions constrained by SCIAMACHY CHOCHO data (Liu et al., 2012b; Alvarado, 2016).

First, we calculate the difference between observed ($C_{CHOCHO}^{SCIAMACHY}$, Fig. 2a) and modeled ($C_{CHOCHO}^{REAM}$, Fig. 2b) CHOCHO VCDs with original emissions ($\Delta C_{CHOCHO} = C_{CHOCHO}^{SCIAMACHY} - C_{CHOCHO}^{REAM}$, Fig. 2c). This discrepancy greatly exceeds the uncertainties of SCIAMACHY retrieval. We then discuss the potential reasons for the difference, i.e. primary emissions from biomass burning and secondary sources from isoprene, acetylene, ethylene and aromatics (Fu et al., 2008; Liu et al., 2012b).

Biomass burning is often a major primary source of CHOCHO (Myriokefalitakis et al., 2008). GFED4.1s inventories, as well as fire hotspots observed by MODIS on board the Terra and Aqua satellites, indicate only a small number of fire occurrences during this period in South Asia, with the exception of crop residue burning in Punjab, an agricultural state in North India. The contribution to CHOCHO VCDs from outdoor biomass burning (Fig. S3a in the Supplement) differs greatly from that of $\Delta C_{CHOCHO}$ (Fig. 2c), which is large over the industrialized Indo-Gangetic Plain. Simulated indoor burning contribution to CHOCHO VCDs is lower by a factor of about 15 than the CHOCHO VCDs discrepancy between satellite retrieval and model simulation (Fig. S3b in the Supplement). The uncertainty of the indoor burning CHOCHO emissions mainly results from that of the emission factor. Even we assume this uncertainty to be 300%, indoor burning cannot explain the low bias of the simulated CHOCHO VCDs. Therefore, the large model underestimation of CHOCHO over the Indo-Gangetic Plain is unlikely due to outdoor biomass burning or indoor burning during our analysis period.

Direct anthropogenic emissions of CHOCHO are small (Volkamer et al., 2005; Stavrakou et al., 2009; Liu et al., 2012b). CHOCHO is produced primarily from the photochemical oxidation of biogenic compounds (e.g., isoprene and terpenes) and hydrocarbon released by anthropogenic activities (e.g., acetylene, ethylene, and aromatics) (Fu et al., 2008). Due to the long atmospheric lifetime of acetylene and ethylene, their contributions to CHOCHO concentrations are quite small in South Asia during October 2010. The most significant secondary sources of CHOCHO in South Asia are isoprene (Fig. 2d) and aromatics (Fig. 2e). Biogenic isoprene emissions depend on vegetation, sunlight, and temperature. The high isoprene contribution to CHOCHO VCDs is to the southeast of the Indo-Gangetic Plain, where CHOCHO VCDs are high in both the observations and model simulations. In comparison, aromatics oxidation dominates CHOCHO over the Indo-Gangetic Plain, where model underestimation is largest (Fig. 2).

We apply the approach by Liu et al. (2012b) to estimate the top-down emissions of aromatics based on SCIAMACHY CHOCHO VCDs. As found by Liu et al. (2012b), domain-wide inversion is impractical since model results correlate poorly with gridded satellite data, most likely reflecting the problems in the spatial distribution of a priori emissions. We therefore determine the emissions by inversion for each grid cell at the overpass time of SCIAMACHY as Liu et al. (2012b) and find similar results for India and nearby regions as Liu et al. (2012b) did for eastern China. The top-down biogenic isoprene

emissions are essentially the same as the a priori emissions. However, the top-down anthropogenic emissions of aromatics (Fig. S4 in the Supplement) increase by a factor of 2-6. The improved model comparison with in situ observations will be discussed in the next section. One caveat with respect to the top-down emission estimate is that we have to assume that the speciation of aromatics in the a priori emission inventory is correct. Since the purpose of this work is to study transport pathways to the Tibetan Plateau on the basis of in situ observations, we examine lumped reactive aromatics (defined as the sum of toluene, ethyl-benzene, and m/p/o-xylene) in the model evaluation (Section 3.2). Satellite observations cannot be used for this purpose since CHOCHO VCDs over Tibet are below or around the detection limit.

## 3 Results and discussion

### 3.1 Observed and simulated reactive aromatics

The average of observed reactive aromatics surface concentration (59±63 pptv) over the Tibetan Plateau is considerably lower than the values found for megacities of China, such as Beijing (8.04 ppbv) and Shanghai (5.2 ppbv) (Liu et al., 2012b). Higher aromatics levels were measured during Period 1 (76±39 pptv) and Period 3 (169±57 pptv) than in Period 2 (26±39 pptv). The model simulation using the a priori emissions in general compares poorly with the in situ observations (Fig. 3). The best performance is during the low-concentration Period 2 when the model underestimates the observations by about a factor of 2. However, the relatively high correlation coefficient ($R^2$=0.83) suggests that atmospheric transport and emission distribution are reasonably simulated. This is in sharp contrast to Periods 1 and 3 when the model underestimates the observations by a factor of 7 with very low correlations between the model and the observations (slope=0.14 and 0.15, $R^2$=0.04 and 0.02 for Period 1 and 3, respectively). We discuss the different reasons for the model performance for Period 1, 2 and 3 in the next 3 sections.

### 3.2 Improvements due to top-down emissions

Figure 2 and Fig. S4 in the Supplement show that SCIAMACHY observations of CHOCHO suggest much higher industrial emissions of aromatics over the Indo-Gangetic Plain than the a priori emissions. We derive top-down emissions on the basis of SCIAMACHY CHOCHO VCDs (Section 2.4). Top-down emissions are higher than the a priori emissions by a factor of 2-6 over the Indo-Gangetic Plain, which is the upwind region of the Tibetan Plateau. Figure 4 shows the resulting improvement in the model simulation. The large underestimations of CHOCHO VCDs over the Indo-Gangetic Plain are corrected as expected (Fig. 4a). At the same time, in situ observations during Period 2 are much better reproduced by the model with the slope increasing from 0.56 to 0.91 and a similar $R^2$ value (0.66) (Fig. 4b). In contrast, model simulations for Periods 1 and 3 are not improved using top-down emission estimates with low biases similar to the original model simulation. This indicates that the reasons for the discrepancies in Periods 1 and 3 are probably not related to the uncertainties in emissions but could be linked to deficiencies in model transport in this area.

## 3.3 Rapid trans-Himalaya transport due to a high-level cut-off Low System

Observed and simulated reactive aromatics concentrations show large variabilities during Period 2 (Fig. 4b). An investigation of these data shows that a major contributor is meteorology. Observed concentrations of reactive aromatics during October 19-20 are generally lower (6.6±3.4 pptv), in comparison to those during October 21-24 (37±45 pptv). The concentration difference during the two time periods is captured by model simulations with top-down emissions (Fig. 5). Analysis of WRF simulated surface wind speed shows an increase by a factor of 2-4 from October 19-20 (Fig. 5a) to 21-24 (Fig. 5b), corresponding well to increasing transport of aromatics from the Indo-Gangetic Plain.

To further analyze the difference between the two time periods, we conduct sensitivity simulations as described in Section 2.3. We compute the source attributions for emissions over Tibet, India and nearby regions, and China excluding Tibet (Fig. 6). During October 19-20, reactive aromatics are due to Tibetan emissions. With the exception of one data point, the concentrations are ≤ 7 pptv. On October 21, emissions from India and nearby regions become dominant while the concentrations are still low (7-21 pptv). During October 22-24, however, emissions from India and nearby regions contribute much higher concentrations (10−137 pptv). The only exception is one data point sampled at 30 km east of Lhasa, where about one fifth of the population of Tibet resides. The contribution by emissions of India and nearby regions to this data point is ~20 pptv, still much higher than during October 19-20. The contribution by emissions from China (excluding Tibet) is negligible (~1%) for this period.

The rise of the Tibetan Plateau is a natural barrier for pollution transport (Fig. 1). Considering the high altitude of the Tibetan Plateau, we analyze 300 hPa geopotential height field in order to understand the change of wind circulation over the region (Fig. 5). During October 19-20, the upper troposphere shows a northward gradual pressure decrease, which does not promote near-surface forcing of trans-Himalaya transport (Fig. 5a). During October 21-24, the presence of a southeastward-moving upper tropospheric cut-off low system induces increasingly stronger surface wind from India to Tibet (Fig. 5b, Hoskins et al., 1985). The cut-off low system is a closed low-pressure system detached from the westerlies. It began to form on October 21 and started to dissipate on October 24. Trans-Himalaya air mass flux in the lower atmosphere shows an increase by a factor of 2 to 5 (Fig. S5 in the Supplement). Accompanying this transport, large amounts of pollutants such as reactive aromatics analyzed here are transported to the Tibetan Plateau leading to much higher surface concentrations.

The cut-off low system provides a more rapid and efficient pollutants transport pathway compared with transport pathways previously proposed by other studies, such as westerlies (Cong et al., 2015; Ji et al., 2015) and mountain-valley winds (Hindman and Upadhyay 2002; Dumka et al., 2010). Compared to aromatics, BC is also subject to wet scavenging, which greatly reduces its transport efficiency by convection. In-cloud BC scavenging is due to cloud activation or ice nucleation and subsequent removal by precipitation, and below-cloud scavenging is due to collision with rain droplets (e.g., Taylor et al., 2014). During our analysis period, the cut-off low system and the associated precipitation are to the northwest of Tibet (Fig. S6 in the Supplement). Precipitation south of Tibet is weak and thus the subsequent removal of BC during trans-Himalaya transport is limited.

To examine the sensitivity of trans-Himalaya transport to the distribution of emission sources, we redistribute the INTEX-B total aromatics emissions over China and other South Asia countries on the basis of the MIX BC emission distributions. We conduct a sensitivity simulation using the redistributed emissions and compared the results to the original simulation. The trans-Himalaya transport from South Asia clearly dominates and it is strongly affected by the presence of a cut-off low system

during our analysis period (Fig. S7 in the Supplement). Our analysis implies that BC transported in the presence of an upper tropospheric cut-off low is potentially a major contributor to BC deposition to Tibetan glaciers.

### 3.4 Missing cut-off low system and complex terrain

Compared to Period 2, model performance for Periods 1 and 3 is very poor with severe low biases (Fig. 3). Transport deficiency appears to be the main problem. Figure 7 shows the histograms of observed and simulated surface wind speed for the 3 periods.

The observed and simulated wind speed distributions are similar for Period 2 (Fig. 7c). In comparison, the simulated wind speed distribution differs drastically for the other two periods (Fig. 7a and 7d).

The wind speed distributions are more similar between Period 1 and 2 in the observations than model simulations. The underestimation of wind speed in Period 1 leads to slower transport of pollutants from the Indo-Gangetic Plain and consequently to a low bias in surface reactive aromatics in the model. Examination of the 300 hPa geopotential height field

during October 13-17 of Period 1 shows a weak trough northeast of Kazakhstan in CSFR reanalysis and WRF simulation results (Fig. S8 in the Supplement). A strong upper tropospheric low pressure system, akin to the cut-off low system of Fig. 5b, will induce stronger lower troposphere wind circulation. The lack of radiosonde observations over the interior of the Tibetan Plateau to constrain the meteorological reanalysis is the plausible reason (Fig. S9 in the Supplement). The horizontal scale of the Rossby Wave at northern mid latitudes is thousands of kilometers, which can be reasonably represented by the

density of the existing radiosonde network. We hypothesize that the smaller scale cut-off low system, not simulated in the reanalysis, is more likely the reason for the model-observation discrepancy during Period 1. We resample surface wind speed of October 23, when a cut-off low system leads to rapid trans-Himalaya transport in Period 2 analyzed in the previous section. At the same time of the day and location as the observations, the simulated wind speed histogram is in good agreement with the observations (Fig. 7b). The corresponding air mass flux across the Himalayas would have been much stronger in the

presence of a cut-off low system (Fig. S10 in the Supplement).

During Period 3, the observed wind speed histogram is skewed to very low wind speed (0-1 m/s) compared to the simulations (Fig. 7d). Sampling bias to avoid locations with strong wind is a possible reason. Another reason is that these samples were taken at lower altitudes in valleys compared to higher altitudes in the other two periods. Inspection of Fig. 1 shows the complex terrain surrounding the valleys of Period 3 sampling. Using high-resolution (~ 1 km) terrain data from the U.S. Geological

Survey (USGS) Global 30 Arc-Second Elevation (GTOPO30) dataset, we find that the standard deviation of altitude in the $7km \times 7km$ region centered at the corresponding observation location correlates well with the observed reactive aromatics with a $R^2$ value of 0.55 (Fig. 8), which suggests that pollution transport is strongly enhanced by the effects of complex terrain. The horizontal resolution of 36 km used in this study is inadequate to simulate this effect. Model resolution as high as 1 km

appears to be necessary to capture the observed feature but the computational resource requirement will be exceptionally large for a global model such as that used for CFSR. Other issues related to complex terrains in this region were also discussed by previous studies (Maussion et al., 2011; Ménégoz et al., 2013; He et al., 2014; Kumar et al., 2015). The effects of complex terrain may have also affected the observations of Periods 1 and 2 but to a smaller extent since the terrain variation is lower
and sampling altitude is higher in those periods.

## 4. Conclusions and implications for climate studies

We apply the REAM model to analyze in situ observations of reactive aromatics across the Tibetan Plateau. Top-down estimate using SCIAMACHY CHOCHO observations suggests that the a priori inventory for aromatics emissions is low by a factor of 2 to 6 over the industrialized Indo-Gangetic Plain. Application of the top-down emission estimate greatly reduces the low bias
of the model during Period 2. Model results suggest that the second half of Period 2 is characterized by rapid trans-Himalaya transport from India and nearby regions driven by the presence of a cut-off low system in the upper troposphere.

Model performance for Periods 1 and 3 is poor compared to Period 2 and employing top-down emission estimates does not significantly improve the model simulation of these periods. In situ observations show much stronger surface wind than simulated in the model during Period 1. The lack of radiosonde observations in the interior of the Tibetan Plateau is likely the
reason that a cut-off low system, the scale of which is much less than the mid-latitude Rossby wave, is not simulated by the T382 (~38 km) CSFR reanalysis. Consequently, trans-Himalaya transport is greatly underestimated in the model. Sampling of Period 3 is in valleys surrounded by complex terrain. Although observed surface wind is weak, we find that reactive aromatics concentrations are strongly correlated with the complexity of surrounding terrain, implying enhanced pollution transport by terrain driven mixing. Model simulations at a resolution of 36 km are inadequate for simulating the terrain effect.

The height of the Tibetan Plateau is a natural barrier for pollution transport into this pristine region. This geographical feature is also a challenge for regional and global model simulations. In this study, we use short-lived reactive aromatics as proxies to evaluate model simulated transport to the Tibetan Plateau on the basis of in situ observations. After correcting for the emission underestimation using satellite observations, simulated trans-Himalaya transport of proxy species (using WRF assimilated meteorological fields) still has significant low biases for two reasons, (1) poor representation of a cut-off low system, and (2)
inadequate representation of terrain effect due to a coarse model resolution. These two transport-related issues likely exist in global climate models; the coarser resolution of climate models than our simulations or CSFR may further worsen the transport biases. Our results imply that pollution transport to the Tibetan Plateau, such as that of BC, is likely to be greatly underestimated in climate models, which was found previously (e.g., He et al., 2014). In addition to trans-Himalaya transport, BC emissions, chemical transformation, and wet deposition also require extensive evaluations with the observations over the
region. Further analysis of reanalysis and climate model simulations is required to quantify potential model biases and the resulting effect of simulated BC deposition to glaciers on the Tibetan Plateau due to the transport issues we identified in this study.

**Acknowledgements**

The modeling analysis of this work was supported by the Atmospheric Chemistry Program of the U.S. National Science Foundation. The observation sampling and analysis were supported by the National Natural Science Foundation of China (No.41472311, 41273107) and the Special Scientific Research Funds for Environmental Protection Commonwealth Section of China (20603020802L). The contributions of the University of Bremen scientists to this manuscript were funded in part by the University and State of Bremen, DLR (German Aerospace), DFG and ESA. MV acknowledges support from the DFG-Research Center / Cluster of Excellence "The Ocean in the Earth System-MARUM". LMAA gratefully acknowledges the funding support by the German Academic Exchange Service (DAAD).

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

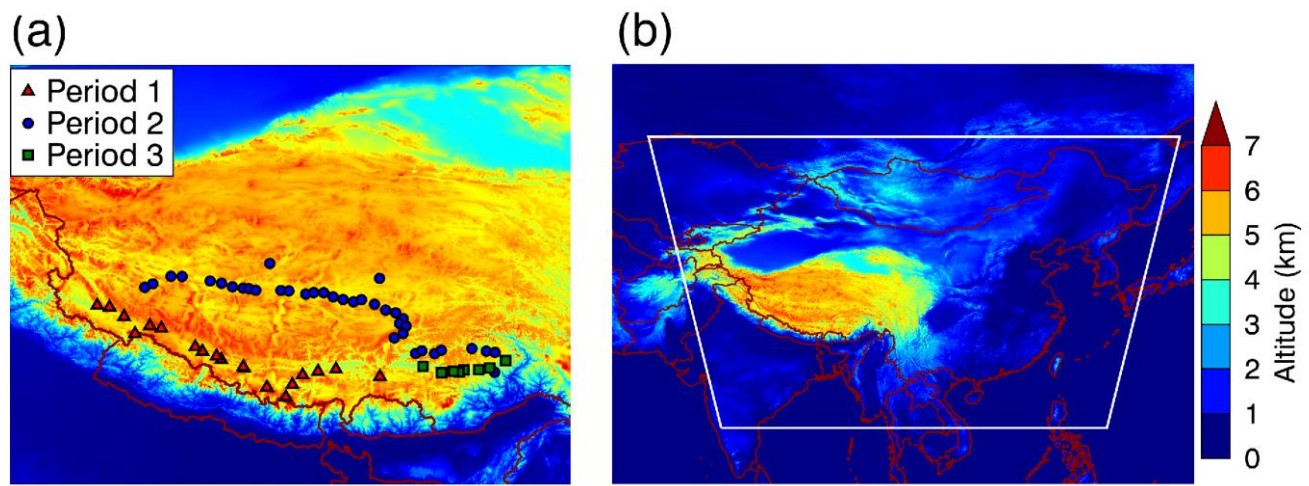

**Figure 1: Overview of regions involved in this study. Locations of observations for Period 1 (October 13-17, 2010, triangle), Period 2 (October 19-24, 2010, circle) and Period 3 (October 25, 2010, square) are shown in (a). White polygon in (b) represents the model domain margin of REAM. Altitude data from Global Topographic Data (GTOPO30, courtesy of the U.S. Geological Survey) is shown as colored background.**

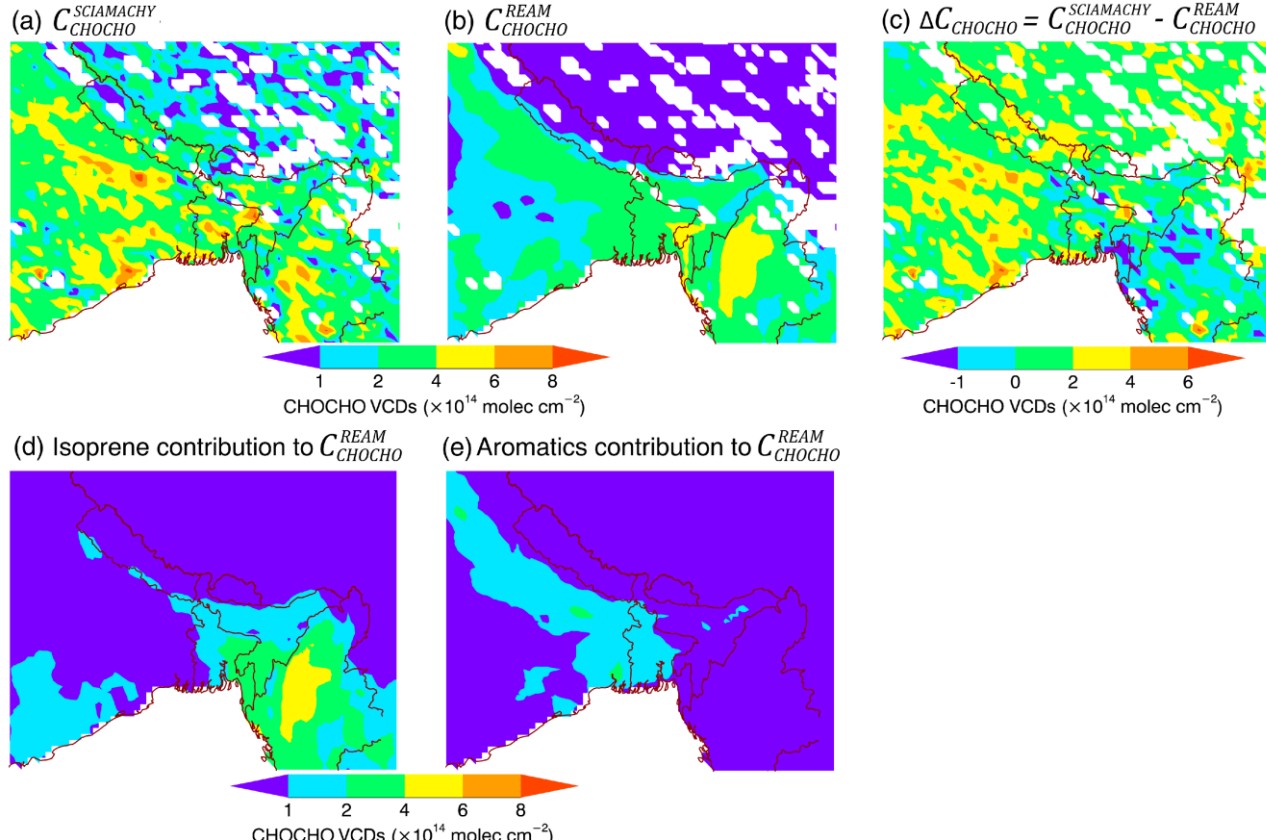

**Figure 2: SCIAMACHY observed CHOCHO VCDs (a), REAM simulated (b) CHOCHO VCDs, the low bias of simulated CHOCHO VCDs (c), simulated isoprene (d) and aromatics (e) contributions to CHOCHO VCDs using the a priori emissions for October 2010. White areas denote missing satellite data or ocean. For each valid SCIAMACHY data point, a corresponding model value is sampled in (b) and (c).**

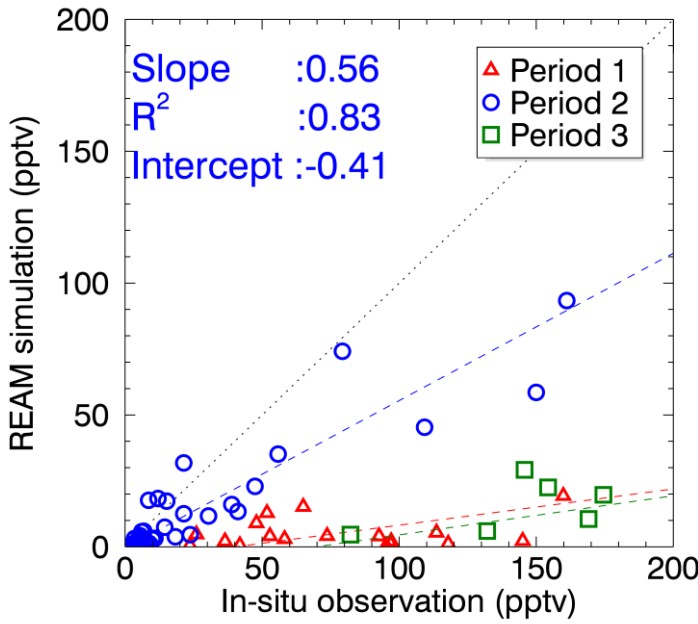

**Figure 3: Comparison between REAM simulated reactive aromatics concentrations (Y-axis) and in situ observations (X-axis). REAM results are archived corresponding to the time and location of the observations. Linear regression results for three periods are shown in red (slope=0.14, R²=0.04), blue (slope=0.56, R²=0.83), and green (slope=0.15, R²=0.02) dashed lines, respectively.**

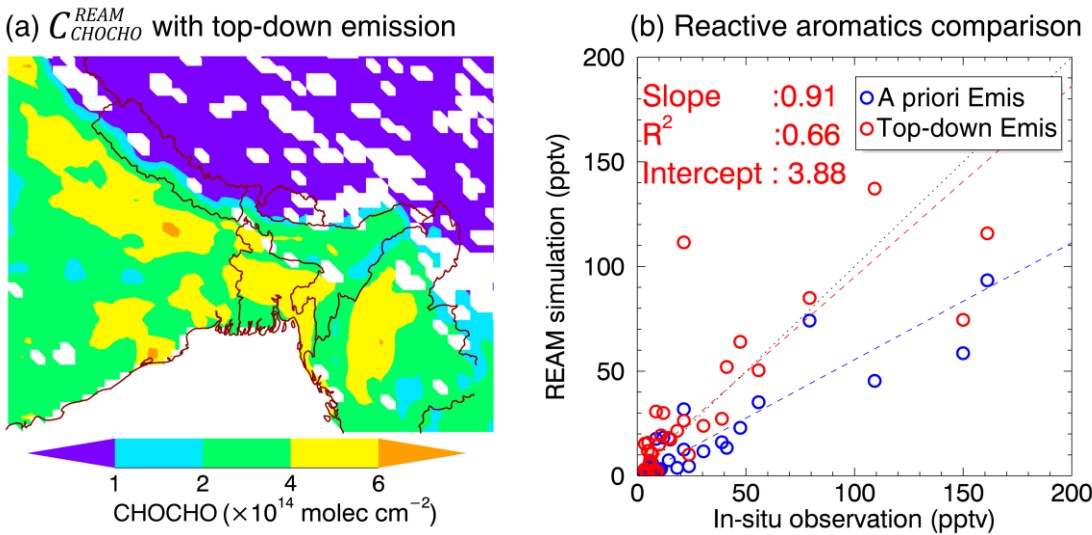

**Figure 4: REAM simulated CHOCHO VCDs with top-down emissions (a) and comparison of simulated and observed reactive aromatics concentrations during Period 2 (b). Blue and black circles in panel (b) represent REAM simulation with a priori (slope=0.56, $R^2$=0.83) and with top-down (slope=0.91, $R^2$=0.66) emissions, respectively.**

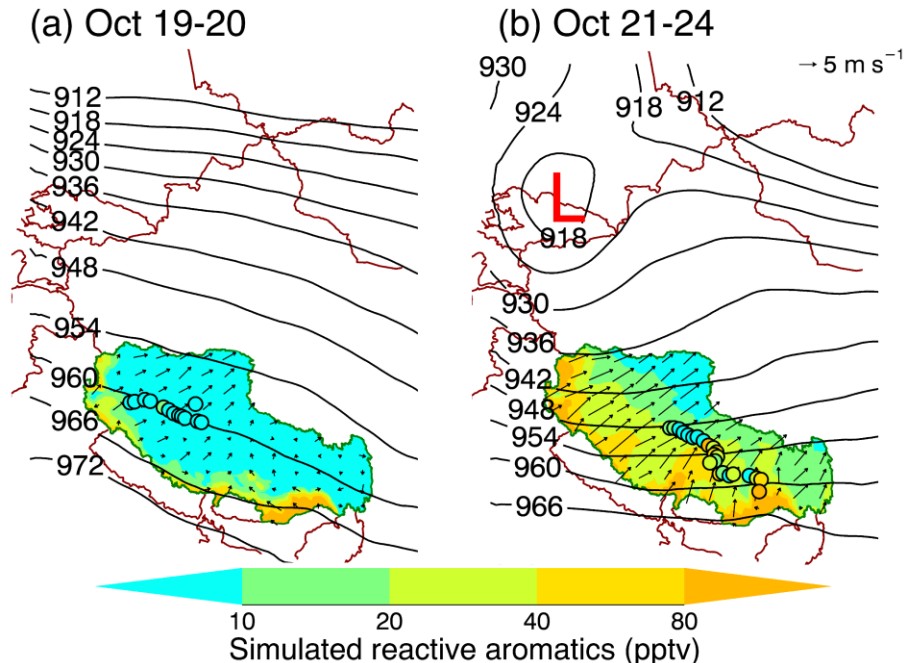

**Figure 5: Distributions of WRF simulated surface wind and REAM simulated concentrations of reactive aromatics over the Tibetan Plateau during October 19-20, 2010 (a) and October 21-24, 2010 (b). Circles show the observed reactive aromatics concentrations. Composite distributions of simulated reactive aromatics concentrations and surface wind over Tibet, corresponding to sampling time of the observations, are shown in color and by arrows, respectively. Corresponding WRF simulated 300 hPa geopotential height fields are shown by contour lines. The border of Tibet Autonomous Region is colored green.**

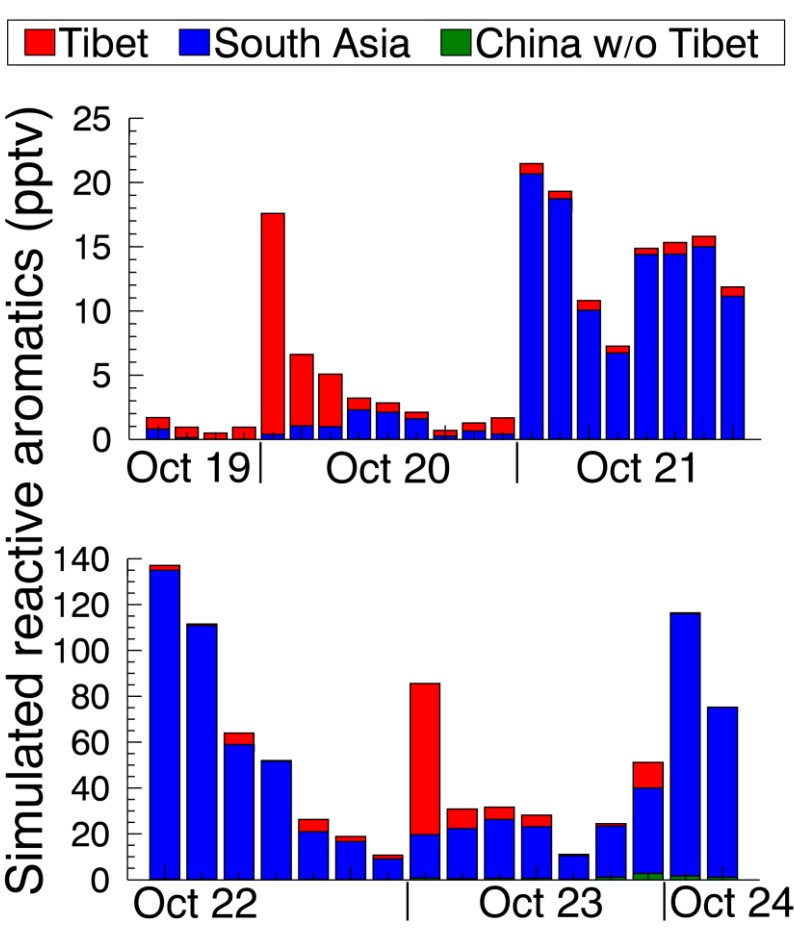

**Figure 6: Reactive aromatics emitted from Tibet (red), India and nearby regions ("South Asia", blue) and China excluding Tibet ('China w/o Tibet', green) corresponding to the in situ observations in the REAM simulation with top-down emissions. Contributions from the other regions are negligible.**

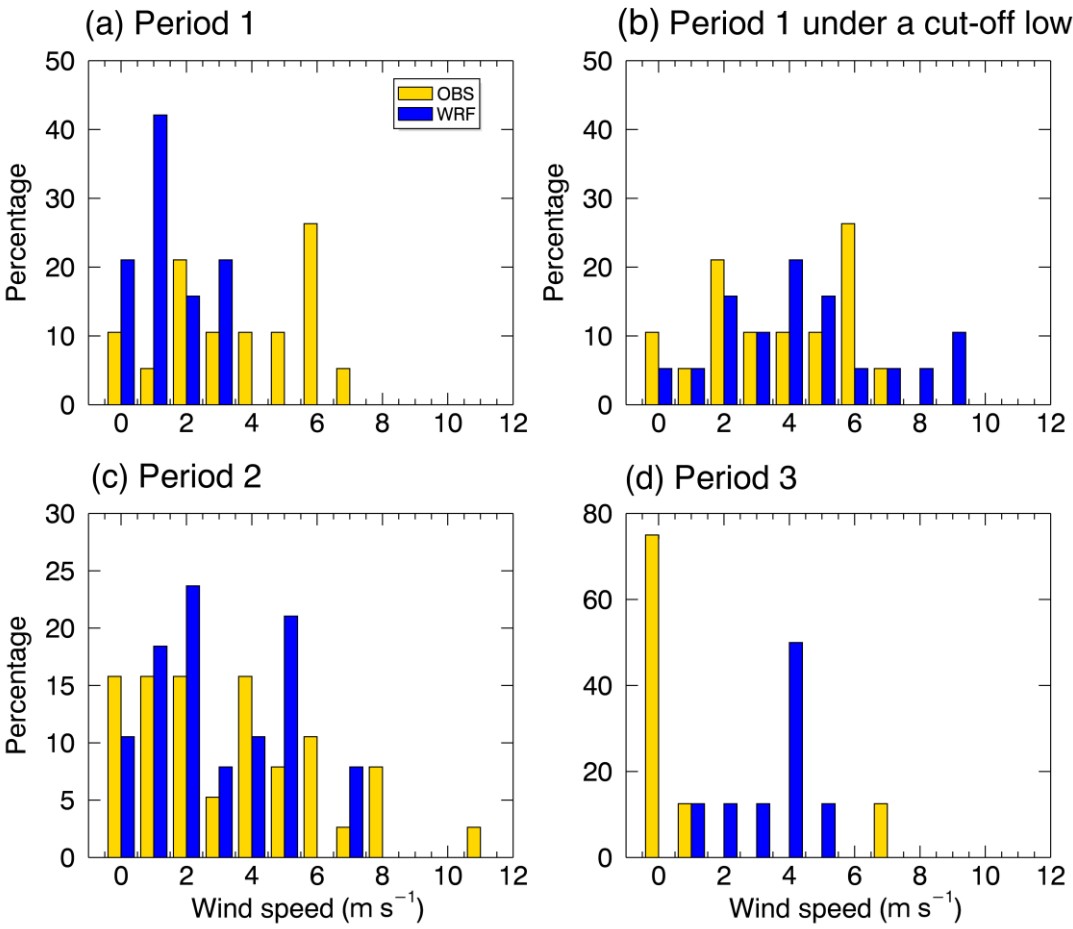

**Figure 7: Histograms of observed and simulated surface wind speed for Period 1 (a), Period 2 (c) and Period 3 (d). Panel (b) shows the wind histogram of October 23 with an upper tropospheric cut-off low system. Model results are sampled at the same time and location as the observations. In Panel (d), the date information is not used. Wind speed is binned at 1 m/s interval.**

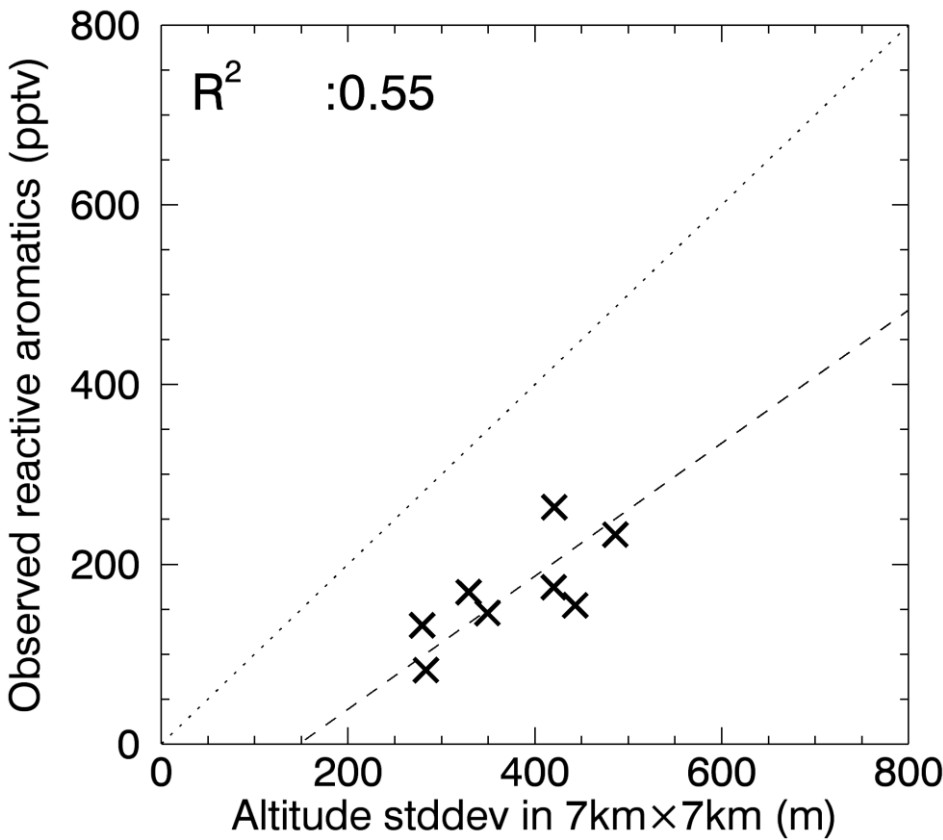

**Figure 8: Observed reactive aromatics as a function of terrain complex during Period 3. The latter is computed as the standard deviation of altitude in a 7km x 7km region centered at the sampling location. The dash line denotes a least-squares regression.**