# Peer review of "Enhanced Trans-Himalaya Pollution Transport to the Tibetan Plateau by the Cut-off Low System"

_Atmospheric Chemistry and Physics, 2016_

## Referee Comment (RC1) · Anonymous Referee #2 · 3 Nov 2016

The paper by Zhang et al. deals with the transport of pollution from the Indo-Gangetic Plain (IGP) to the Tibetan Plateau. The Authors provide a brief review of the experimental findings from in situ observations. Then, they present the results of a regional transport model in simulating the concentrations of anthropogenic aromatic hydrocarbons (HCs) measured in Tibet in Oct 2010. Finally, they present an approach for correcting possible biases in current emission inventories, and highlight possible problems with state-of-the-art models in simulating transport events associated with specific meteorological conditions. The paper is a nice, clear exercise of observation-constrained modeling. I have one major comment about the impact of this study and two additional specific comments.

First of all, the Authors claim that their analysis has implications for improving the modelling of black carbon (BC) transport to the glaciated regions of Tibet (all the Introduction is dedicated to this topic). However, their approach is based on measurements (in situ and satellite retrievals) of aromatic HCs and of their degradation products (glyoxal). It is certainly true that aromatic hydrocarbons share with BC several emission and transport patterns, but only to a certain extent. For instance, the aromatic HCs are emitted by fossil fuel combustion, gasoline evaporation and solvent use (page 2, line 24), however only the first of these three sectors is of importance for BC. It follows that top-down methods for correcting the emissions of aromatic HCs (Section 2.4) has unclear implications for improving the representation of BC sources in the models. If the scope of the paper is really improving BC modelling in the Himalayan-Tibetan region, then the absence of BC observations poses a major caveat, even if the approach is conceptually valid and in principle it could be extended to experiments involving real BC measurements.

Specific comments:

a. Biomass burning is ruled out from the possible explanations for the difference between observed and retrieved glyoxal concentrations over the IGP, because satellite fire counts show only spot fire occurrence over the Plain with little correspondence with the model-measurement gap (Page 5, lines 10 – 14). However, open burning accounts for only a fraction of biomass burning, which is normally practiced also indoor for cooking, heating etc., undetected by remote sensing. Therefore, I would not rule out the hypothesis of a direct emission of glyoxal from domestic biomass burning.

b. The Authors find a plausible explanation for the rise of aromatic HCs concentrations between 22 and 24 Oct 2010 in the synoptic meteorological conditions in central Asia showing an upper-level cut-off system triggering a southerly circulation from India to Tibet. However, minding that BC can be removed during transport by precipitations, the Authors should provide a more in-depth analysis of the meteorological conditions over the Himalayans during the approach of the low-pressure system. Ap-

parently, on the 22 of October, frontal cloud systems travelled over the Tibet from west to east (http://www.ssec.wisc.edu/data/comp/ir/2010295M0000.gif). The presence of precipitations with possible losses of BC (and not necessarily of aromatic HCs) in the Himalayas should be checked carefully at local meteorological stations.

---

## Referee Comment (RC2) · Anonymous Referee #1 · 10 Nov 2016

Summary:

Zhang et al. use the REAM chemistry transport model simulation to investigate transport of aromatics to the Tibetan plateau. Their work shows that the INTEX-B 2006 emissions of aromatics do not produce sufficient glyoxal concentrations compared to the SCIAMACHY retrieval. The authors apply a top-down estimate to update the emissions of aromatics, which are glyoxal precursors. The REAM model results of aromatics were compared with observations taken at several ground locations over a 3-week period. Samples in central Tibet had the highest aromatic concentrations and were attributed to meteorological conditions that increased southwesterly surface winds bringing high concentrations of aromatics from the Indo-Gangetic Plain to Tibet.

[Figure]

The complex topography of this region makes for an especially challenging effort to represent transport into Tibet.

The investigation is important in terms of understanding transport of pollutants, especially black carbon, from population and industrial regions to the Himalayan glaciers. The results from this paper suggest the critical need to represent the airflow in complex terrain to predict black carbon transport accurately. While these conclusions are not unfamiliar, it is important to continue to highlight the role of meteorology on transport of pollutants. The presentation of the investigation is fair. One can understand the points being made, but it is not written as a compelling story. Several of the points below suggest ways to improve the paper.

Major Comments

1. Aromatics are good markers of transport that occurs over $\sim$1 day period because of their chemical lifetime. However, aromatics are not subject to wet deposition because of their low solubilities (Sander, 2015), while black carbon can be removed by storms. Therefore, it makes sense to use aromatics to analyze transport (isolating the one process), but they are not good proxies for black carbon. The authors should explain this caveat in the paper.

2. There is a lack of recognition of previous studies, especially of regional chemistry transport modeling in South Asia and western China. Some previous papers to consider are listed in the references.

3. When figures are discussed in enough detail, it is better to place them in the main part of the paper. In my opinion, the supplement should not contain information that is needed to support the conclusions of the paper. For example, Figure S2 should be part of the main paper because it supports the conclusion that the INTEX-A aromatics emission estimates are much lower than values determined from a top-down estimate. Please write the paper so that the reader can easily understand the main points of the study.

Minor Comments:

1. Page 4, Line 1: How does one choose $\alpha$ for use in the uncertainty equation? Is this even necessary information for the reader?

2. I suggest rearranging the model description paragraphs. How would you describe the model to someone who has never worked with regional transport models? I suggest beginning with the CFSR dataset because it is used to provide initial and boundary conditions for the WRF model. Then the WRF model should be described, citing what version, resolution, and parameterizations are used. Next, it should be stated how REAM takes information from the WRF simulation. Does it take WRF output every hour, every 3 hours, etc.? Finally, the REAM model should be described. Do not rely on the reader to go to the cited references to get needed information, but instead to go to the cited references to get more details.

3. It is important to include what the model top is because of the high surface elevation of Tibet that is prone to have stratospheric intrusions (perhaps falsely if the model top is too low).

4. Has the REAM model been evaluated for the region simulated? In this paper we see comparisons with SCIAMACHY and ground-based observations. How does the model perform in terms of meteorology and chemical constituents, such as CO, O3, NOx, and particulate matter?

5. The model domain is shown in Figure 1, allowing the reader to recognize that the region of interest is mostly away from the model boundaries. Is the Tibet region affected by the composition outside the model boundaries (especially the western boundary), or outside the subdomain shown in Figure 2?

6. When comparing the REAM model results with the SCIAMACHY satellite retrieval of glyoxal, is the model sampled the same way as SCIAMACHY sees the atmosphere? For example, I assume that the missing data in Figure 2a from the satellite data is due

to clouds. Are cloudy grid points removed from the model analysis (it doesn't appear so since there are no "missing data" from the model results).

7. Page 5, Line 21. It would be helpful to see a MEGAN emissions map of isoprene for the region.

8. Page 5, Lines 16-24. It may be helpful to include the glyoxal chemistry in the supplement.

9. Section 2.4. Why is the INTEX-B emissions inventory, which is appropriate for year 2006, being used for the model simulation for year 2010? MACCity emissions (appropriate for 2010) or EDGAR-HTAP emissions may have been better suited for these simulations. Could the authors discuss the differences between the emissions inventory that they used and these more recent emissions inventories?

10. Section 3.1. It would be interesting to learn in more detail what the surface elevation is at the observation points and at the matching REAM model grid cells. Could there be discrepancies between model and observations because the model does not adequately represent the surface elevation?

11. Page 7, Line 6. How are the source attributions computed?

12. Page 7, Lines 15-17. Could the "cutoff low system" be described in more detail? Would "closed low" be a more appropriate term? (see the NWS definition at http://forecast.weather.gov/glossary.php?word=cutoff%20low) How long did the cutoff low remain in the region? Was there precipitation associated with the cutoff low?

13. Page 7-8. It would be helpful to see Figure S5 showing both Period 1 and Period 2. From what is presented, it is unclear whether WRF simulates the cutoff low pressure system (unless these are WRF results, which is not clear from the figure caption).

14. Page 8, End of section 3. There should be a section added, discussing the results found in this study with previous papers (such as those listed in the references). For example, the Kumar et al. (2015) study also mentions the challenges of modeling

pollutant transport in the Himalayas. Ji et al. (2015) also discuss aerosol transport from the IGP to Tibet.

15. Figure 5. What is the source of the information plotted in Figure 5? Is it from the model (WRF + REAM) simulation? Please clarify. Why are the surface winds and simulated reactive aromatics shown only for Tibet?

Technical Comments:

1. Page 1, Line 16: verb agreement: Long-range transport followed by deposition of black carbon on glaciers of Tibet is one of the key issues

2. Page 1, Line 17: → impacting the melting of glaciers

3. Page 1, Line 28: Remove "Furthermore"

4. Page 2, Line 2: The Menon et al. (2002) paper should be cited.

5. Page 2, Line 5: Insert "altitude" or "surface elevation" after "4 km"

6. Page 2, Line 14 is an orphan sentence and is not really needed.

7. Page 2, Line 19: "aerosols" may be a better word than "condensation nuclei"

8. Page 2, Line 20: "large-scale westerlies from East Asia" does not make sense. East Asia is east of Tibet, so it must be an easterly wind if the air moves east to west.

9. Page 3, Line 20: Shouldn't Fig. 1a be cited before Fig. 1b?

10. Page 3, Line 23: I think it should be "overpass time" and not "overpassing".

11. Page 4, Lines 4-8: Are all these references needed? It is sufficient to just cite 1-2 example references per topic.

12. Page 4, Line 24: It would be good to cite Figure S1a.

13. Page 5, Lines 2-4 is a long sentence. Please break it up into 2 sentences.

14. Page 5, Line 2: I think it should be "overpass time" and not "overpassing".

15. Page 5, Line 28: I think it should be "overpass time" and not "overpassing".

16. Page 5, Line 29: Insert "did" before "for eastern China".

17. Page 7, Line 15: I would suggest using "promote" instead of "provide".

18. Page 7, Line 24: Are the histograms for wind speed for at the surface (or 10-m winds)? Please clarify.

19. References: Could the references be written so that they are easier to read? Either adding a "hanging indent" or a line space between references would help immensely.

20. Figure 4b: The black and dark blue colors are quite similar. Could a different color be plotted?

21. Figure S3: To emphasize the differences between the panels, it may be better to plot using the same scaling. The gradients can still be appreciated if a "log type" scaling is used, e.g. 1, 2, 3, 5, 7, 10.

22. Figure S7: The legend mistypes "original". The original winds line does not look like the black line in Figure S4d.

References

These are mostly for BC studies for South Asia region, but do include other papers cited above.

Dumka, U. C., et al. (2010), Characteristics of aerosol black carbon mass concentration over a high altitude location in the Central Himalayas from multi-year measurements, Atmos. Res., 96 (4), 510–521.

Guha, A., et al. (2015), Seasonal characteristics of Aerosol Black carbon in relation to Long Range transport over Tripura in Northeast India, Aerosol and Air Quality Research, Aerosol and Air Quality Research, 15: 786–798, doi:

10.4209/aaqr.2014.02.0029.

Ji, Z., S. Kang, Z. Cong, Q. Zhang, and T. Yao, (2015) Simulation of carbonaceous aerosols over the Third Pole and adjacent regions: distribution, transportation, deposition, and climatic effects, Clim. Dyn., 45, 2831-2846, doi:10.1007/s00382-015-2509-1

Koch, D. and A. D. Del Genio, (2010), Black carbon semi-direct effects on cloud cover: review and synthesis, Atmos. Chem. Phys., 10, 7685-7696, doi:10.5194/acp-10-7685-2010.

Kumar, R., M. C. Barth, G. G. Pfister, V. S. Nair, S. D. Ghude, and N. Ojha (2015), What controls the seasonal cycle of black carbon aerosols in India?. J. Geophys. Res. Atmos., 120, 7788–7812. doi: 10.1002/2015JD023298.

Lau, K. M., M. K. Kim, and K. M. Kim, (2006), Asian summer monsoon anomalies induced by aerosol direct forcing: The role of the Tibetan Plateau, Clim. Dyn., 26, 855–864, doi:10.1007/s00382-006-0114-z.

Lawrence, M. G. and J., Lelieveld, (2010), Atmospheric pollutants outflow from southern Asia: a review, Atmos. Chem. Phys., 10, 11017-11096, doi: 10.5194/acp-10-11017-2010. Menon, S., J. Hansen, L. K. Nazaren, and Y. Leo, (2002), Climate effects of BC aerosols in China and India, Science, 297 (5590), 2250–2253.

Nair, V. S., et al., (2012), Simulation of South Asian aerosols for regional climate studies, J. Geophys. Res., 117, D04209, doi: 10.1029/2011JD016711, 2012.

Nair, V. S., et al., (2013), Black carbon aerosols over the Himalayas: direct and surface albedo forcing, Tellus B, 65, 19738, doi: 10.3402/tellusb.v65i0.19738.

Ramanathan, V., and G. Carmichael, (2008), Global and regional climate changes due to black carbon, Nature Geosci., 1, 221-227.

Sander, R. (2015) Compilation of Henry's law constants (version 4.0) for water as solvent, Atmos. Chem. Phys., 15, 4399-4981, doi:10.5194/acp-15-4399-2015.

Yasunari, T. J., et al., (2010), Estimated impact of black carbon deposition during pre‐monsoon season from Nepal Climate Observatory-Pyramid data and snow albedo changes over Himalayan glaciers, Atmos. Chem. Phys., 10, 6603–6615, doi:10.5194/acp-10-6603-2010.

---

## Author Comment (AC1) · 14 Nov 2016

**Initial Response to Referee #2:**

We thank the reviewer for the helpful comments. Here we will provide quick responses to these comments for the purpose of interactive discussion and will update the paper text and supplement after the completion of the interactive discussion.

*First of all, the Authors claim that their analysis has implications for improving the modelling of black carbon (BC) transport to the glaciated regions of Tibet (all the Introduction is dedicated to this topic). However, their approach is based on measurements (in situ and satellite retrievals) of aromatic HCs and of their degradation products (glyoxal). It is certainly true that aromatic hydrocarbons share with BC several emission and transport patterns, but only to a certain extent. For instance, the aromatic HCs are emitted by fossil fuel combustion, gasoline evaporation and solvent use (page 2, line 24), however only the first of these three sectors is of importance for BC. It follows that top-down methods for correcting the emissions of aromatic HCs (Section 2.4) has unclear implications for improving the representation of BC sources in the models. If the scope of the paper is really improving BC modelling in the Himalayan-Tibetan region, then the absence of BC observations poses a major caveat, even if the approach is conceptually valid and in principle it could be extended to experiments involving real BC measurements.*

**Authors' response:**

**The implications of this paper for BC are only on transport. As the reviewer pointed out that our work cannot directly address the accuracy of BC emission inventories, which we did not claim in the paper that we can either. We suggest that aromatics observations are good proxies for understanding transport processes of BC to the Tibetan plateau. The major pathway of transport is driven by the presence of a cut-off low system. Section 2.4 shows that the underestimates of aromatics transport are due to an underestimation of emissions, which can be improved using satellite observations.**

**To further demonstrate the link between transport of aromatics and BC to Tibet, we redistributed total aromatics emissions over China and other South Asia countries using the BC emission distributions. Therefore, the resulting aromatics emission distributions resembles that of BC. We conducted a sensitivity simulation using these emissions and compared the results to the original REAM simulation (Fig. S8). Transport of BC from South Asia (e.g., India) clearly dominates and it is strongly affected by the presence of a cut-off low system, as we discussed in the paper.**

[Figure]

**Figure S8: Averages of simulated reactive aromatics emitted from Tibet (red), India and nearby countries ("South Asia", blue) and China excluding Tibet ("China w/o Tibet", green) corresponding to in situ observations during Oct 19-20 and Oct 21-24. REAM simulations are conducted with original emissions (a) and the aromatics emissions redistributed to be the same as BC (b), respectively.**

*Specific comments: a. Biomass burning is ruled out from the possible explanations for the difference between observed and retrieved glyoxal concentrations over the IGP, because satellite fire counts show only spot fire occurrence over the Plain with little correspondence with the model-measurement gap (Page 5, lines 10 – 14). However, open burning accounts for only a fraction of biomass burning, which is normally practiced also indoor for cooking, heating etc., undetected by remote sensing. Therefore, I would not rule out the hypothesis of a direct emission of glyoxal from domestic biomass burning.*

**We calculate indoor biomass burning glyoxal emissions using emission factors from Pettersson et al. (2011) and Li et al. (2014). The Indian rural and urban population distribution from the NASA Socioeconomic Data and Applications Center (SEDAC) Network for year 2010 is used as spatial proxies. We adopt the energy consumptions for rural and urban inhabitants on the basis of the National Sample Survey Office of India (N.S.S.O., 2012a, 2012b). Simulated indoor biomass burning contribution to glyoxal VCDs is lower by a factor of about 15 than the glyoxal VCDs discrepancy between satellite retrieval and model simulation (Fig. 1). The uncertainty of the indoor energy consumption glyoxal emissions mainly results from the uncertainty of the emission factor. Even we assume this uncertainty to be 300%, indoor burning cannot explain the low bias of the simulated glyoxal VCDs. Thus, we consider that the underestimation of aromatics emissions is the main reason for the glyoxal VCD discrepancy.**

[Figure]

**Figure 1: Simulated contribution of indoor burning to CHOCHO VCDs.**

*b. The Authors find a plausible explanation for the rise of aromatic HCs concentrations between 22 and 24 Oct 2010 in the synoptic meteorological conditions in central Asia showing an upper-level cut-off system triggering a southerly circulation from India to Tibet. However, minding that BC can be removed during transport by precipitations, the Authors should provide a more in-depth analysis of the meteorological conditions over the Himalayans during the approach of the low-pressure system. Apparently, on the 22 of October, frontal cloud systems travelled over the Tibet from west to east (http://www.ssec.wisc.edu/data/comp/ir/2010295M0000.gif). The presence of precipitations with possible losses of BC (and not necessarily of aromatic HCs) in the Himalayas should be checked carefully at local meteorological stations.*

**This is a good point for BC concentration distribution over the Tibet, which is not studied in this work. Our focus is on transport. BC still needs to be transported to the Tibetan region; scavenging can only reduce the BC amount. If BC is not transported to the region in the first place, scavenging cannot increase BC in this region.**

**Our study suggests that the cut-off low system is difficult to simulate with a regional model constrained by meteorological observations. It would be much harder for climate models to simulate correctly. Fig. 2 shows the precipitation distribution for Oct 19-20 and Oct 21-24. The cut-off system is to the northwest of Tibet. Precipitation south of the Himalayas is weak and hence the removed BC from precipitation during the inter-Himalayas transport is limited.**

[Figure]

**Figure 2: WRF simulated averaged daily precipitation for Oct 19-20 (a) and Oct 21-24 (b), respectively.**

**References:**

Center for International Earth Science Information Network - CIESIN - Columbia University, International Food Policy Research Institute - IFPRI, The World Bank, and Centro Internacional de Agricultura Tropical - CIAT: Global Rural-Urban Mapping Project, Version 1 (GRUMPv1): Population Count Grid, in, NASA Socioeconomic Data and Applications Center (SEDAC), Palisades, NY, 2011.

Center for International Earth Science Information Network - CIESIN - Columbia University: Gridded Population of the World, Version 4 (GPWv4): Population Density Adjusted to Match 2015 Revision UN WPP Country Totals, in, NASA Socioeconomic Data and Applications Center (SEDAC), Palisades, NY, 2016.

Li, M., Zhang, Q., Streets, D. G., He, K. B., Cheng, Y. F., Emmons, L. K., Huo, H., Kang, S. C., Lu, Z., Shao, M., Su, H., Yu, X., and Zhang, Y.: Mapping Asian anthropogenic emissions of non-methane volatile organic compounds to multiple chemical mechanisms, Atmos. Chem. Phys., 14, 5617-5638, 10.5194/acp-14-5617-2014, 2014.

N. S. S. O. (NSSO), Household Consumption of various Goods and Services in India, (July 2009-June 2010), vol. KI of 69th round. National Sample Survey Office, Ministry of Statistics & Programme Implementation, Government of India, 2012a

N. S. S. O. (NSSO), Energy Sources of Indian Households, (July 2009-June 2010), vol. KI of 69th round. National Sample Survey Office, Ministry of Statistics & Programme Implementation, Government of India, 2012 b

Pettersson, E., Boman, C., Westerholm, R., Boström, D., and Nordin, A.: Stove Performance and Emission Characteristics in Residential Wood Log and Pellet Combustion, Part 2: Wood Stove, Energy & Fuels, 25, 315-323, 10.1021/ef1007787, 2011.

---

## Author Comment (AC2) · 19 Nov 2016

We thank the reviewer for the thoughtful and detailed comments. Here we will provide quick responses only to the major comments for the purpose of interactive discussion. We will respond to the other comments and update the paper text and supplement after the completion of the interactive discussion.

Please also note the supplement to this comment: http://www.atmos-chem-phys-discuss.net/acp-2016-702/acp-2016-702-AC2-supplement.pdf

[Figure]

**Supplement:**

**Initial Response to Referee #1:**

We thank the reviewer for the thoughtful and detailed comments. Here we will provide quick responses *only* to the major comments for the purpose of interactive discussion. We will respond to the other comments and update the paper text and supplement after the completion of the interactive discussion.

*Summary:*
*Zhang et al. use the REAM chemistry transport model simulation to investigate transport of aromatics to the Tibetan plateau. Their work shows that the INTEX-B 2006 emissions of aromatics do not produce sufficient glyoxal concentrations compared to the SCIAMACHY retrieval. The authors apply a top-down estimate to update the emissions of aromatics, which are glyoxal precursors. The REAM model results of aromatics were compared with observations taken at several ground locations over a 3-week period. Samples in central Tibet had the highest aromatic concentrations and were attributed to meteorological conditions that increased southwesterly surface winds bringing high concentrations of aromatics from the Indo-Gangetic Plain to Tibet. The complex topography of this region makes for an especially challenging effort to represent transport into Tibet.*
*The investigation is important in terms of understanding transport of pollutants, especially black carbon, from population and industrial regions to the Himalayan glaciers. The results from this paper suggest the critical need to represent the airflow in complex terrain to predict black carbon transport accurately. While these conclusions are not unfamiliar, it is important to continue to highlight the role of meteorology on transport of pollutants. The presentation of the investigation is fair. One can understand the points being made, but it is not written as a compelling story. Several of the points below suggest ways to improve the paper.*

**Authors' response:**
**A main focus of this research is to show the enhanced inter-Himalayas transport towards Tibet in the presence of the cut-off low system. We apply the top-down emission estimate technique and find that the model underestimates of reactive aromatics during Period 2 result due to the underestimation of the emissions inventory. The WRF simulated and observed meteorology fields show that the rise of reactive aromatics concentrations during Oct 21-24 are related to the cut-off low system. We further analyze the geopotential heights and compare WRF simulated and observed surface winds during Period 1. The results imply a missing cut-off low system during Period 1, which is not captured by 36km resolution WRF model. It will be more challenging for climate models with coarser resolutions to capture the cut-off low systems. Thus, we suggest climate model results of inter-Himalayas transport of pollutants (including BC) must be evaluated with our findings in mind. Last, we point out that the model underestimation during Period 3 is due to the effects of complex terrains.**

*Major Comments*
*1. Aromatics are good markers of transport that occurs over 1 day period because of their chemical lifetime. However, aromatics are not subject to wet deposition because of their low solubilities (Sander, 2015), while black carbon can be removed by storms. Therefore, it makes sense to use aromatics to analyze transport (isolating the one process), but they are not good proxies for black carbon. The authors should explain this caveat in the paper.*

**Although air mass transport towards north of the Himalayas is enhanced during the presence of the cut-off low system, different species share various physical and chemical processes and are affected by the cut-off low system to different extents. Compared to aromatics, black carbon (BC) transport is also subject to wet scavenging, the efficiency of which depends on the hygroscopicity of BC. For freshly emitted BC from the industrialized Indo-Gangetic Plain (IGP), the scavenging effect requires in situ observation. Nonetheless, inter-Himalayas transport during Period 2 is clearly much faster than Period 1. If this process is not simulated correctly in the model, BC transport from India to Tibet will likely be underestimated. Further, Fig. 1 shows that the precipitation distribution for Oct 19-20 and Oct 21-24. The cut-off system and associated precipitation are to the northwest of Tibet. Precipitation south of Tibet is weak and thus the subsequent removal of BC during inter-Himalayas transport is limited.**

[Figure]

**Figure 1: WRF simulated averaged daily precipitation for Oct 19-20 (a) and Oct 21-24 (b), respectively.**

In addition, we conducted a sensitivity test in order to address the influence of different emission distributions between aromatics and BC to our conclusion. We redistributed total aromatics emissions over China and other South Asia countries using the BC emission distribution. The resulting aromatics emissions distribution resembles that of BC. We compare the source contribution results using original and the redistributed aromatics emissions (Fig. S8). Transport of BC from South Asia clearly dominates inner Tibet during the cut-off low event.

Thus, we conclude that the BC transport is enhanced by the cut-off low. We will clarify these issues in the revision.

[Figure]

**Figure S8: Averages of simulated reactive aromatics emitted from Tibet (red), India and nearby countries ("South Asia", blue) and China excluding Tibet ("China w/o Tibet", green) corresponding to in situ observations during Oct 19-20 and Oct 21-24. REAM simulations are conducted with original emissions (a) and the aromatics emissions redistributed to be the same as BC (b), respectively.**

*2. There is a lack of recognition of previous studies, especially of regional chemistry transport modeling in South Asia and western China. Some previous papers to consider are listed in the references.*

**We will do another search of literature. Our research focuses on the inter-Himalayas transport from South Asia to Tibet. We will check if there are papers on inter-Himalayas transport that we missed.**

*3. When figures are discussed in enough detail, it is better to place them in the main part of the paper. In my opinion, the supplement should not contain information that is needed to support the conclusions of the paper. For example, Figure S2 should be part of the main paper because it supports the conclusion that the INTEX-A aromatics emission estimates are much lower than values determined from a top-down estimate. Please write the paper so that the reader can easily understand the main points of the study.*

**Thanks for the suggestion. We will combine Fig. S2 and Fig. 2 in the main text to support our top-down estimate. Please note that we did not use INTEX-A emission estimates in this work. Only INTEX-B data are used.**

---

## Author Response (AR1)

We appreciate both reviewers for their helpful comments and suggestions concerning our research. We revised the manuscript accordingly. We present our response and changes below. Reviewers' comments and suggestions are shown in *italic*. Authors' responses are in **bold**.

**Response to Referee #1:**

*Summary:*
*Zhang et al. use the REAM chemistry transport model simulation to investigate transport of aromatics to the Tibetan plateau. Their work shows that the INTEX-B 2006 emissions of aromatics do not produce sufficient glyoxal concentrations compared to the SCIAMACHY retrieval. The authors apply a top-down estimate to update the emissions of aromatics, which are glyoxal precursors. The REAM model results of aromatics were compared with observations taken at several ground locations over a 3-week period. Samples in central Tibet had the highest aromatic concentrations and were attributed to meteorological conditions that increased southwesterly surface winds bringing high concentrations of aromatics from the Indo-Gangetic Plain to Tibet. The complex topography of this region makes for an especially challenging effort to represent transport into Tibet.*
*The investigation is important in terms of understanding transport of pollutants, especially black carbon, from population and industrial regions to the Himalayan glaciers. The results from this paper suggest the critical need to represent the airflow in complex terrain to predict black carbon transport accurately. While these conclusions are not unfamiliar, it is important to continue to highlight the role of meteorology on transport of pollutants. The presentation of the investigation is fair. One can understand the points being made, but it is not written as a compelling story. Several of the points below suggest ways to improve the paper.*

**A main focus of this research is to show the enhanced trans-Himalaya transport towards Tibet in the presence of the cut-off low system. We apply the top-down emission estimate technique and find that the model underestimates of reactive aromatics during Period 2 result from the underestimation of the emission inventory. The WRF simulated and the observed meteorology fields show that the rise of reactive aromatics concentrations during October 21-24 is related to the cut-off low system. We further analyze the geopotential heights and compare WRF simulated and observed surface winds during Period 1. The results imply a missing cut-off low system during Period 1, which is not captured by the 36km resolution WRF model. It will be more challenging for climate models with coarser resolutions to capture the cut-off low systems. Thus, we suggest climate model results of trans-Himalaya transport of pollutants (including BC) must be evaluated with our findings in mind. Last, we point out that the model underestimation during Period 3 is due to the effects of complex terrains.**

*Major Comments*
*1. Aromatics are good markers of transport that occurs over 1 day period because of their chemical lifetime. However, aromatics are not subject to wet deposition because of their low solubilities (Sander, 2015), while black carbon can be removed by storms. Therefore, it makes sense to use aromatics to analyze transport (isolating the one process), but they are not good proxies for black carbon. The authors should explain this caveat in the paper.*

**Although air mass transport towards north of the Himalayas is enhanced during the presence of the cut-off low system, different species share various physical and chemical processes and are affected by the cut-off low system to different extents. Compared to aromatics, black carbon (BC) transport is also subject to wet scavenging, the efficiency of which depends on the hygroscopicity of BC. For freshly emitted BC from the industrialized Indo-Gangetic Plain (IGP), the scavenging effect requires in situ**

observations. Nonetheless, trans-Himalaya transport during October 21-24 is clearly much faster than that during October 19-20. If this process is not simulated correctly in the model, BC transport from India to Tibet will likely be underestimated. Further, Fig. S6 in the Supplement shows that the precipitation distribution for October 19-20 and October 21-24. The cut-off low system and associated precipitation are to the northwest of Tibet. Precipitation south of Tibet is weak and thus the subsequent removal of BC during trans-Himalaya transport is limited.

[Figure]

**Figure S6: WRF simulated averaged daily precipitation for October 19-20 (a) and October 21-24 (b), respectively.**

In addition, we conducted a sensitivity simulation in order to address the sensitivity of our modeling results to the emission distribution difference between aromatics and BC. We redistributed the total aromatics emissions over China and other South Asia countries on the basis of the BC emission distribution. The resulting aromatics emissions distribution resembles that of BC. We compare the source contribution results using original and the redistributed aromatics emissions (Fig. S7). Transport of BC from South Asia clearly dominates over Tibet during the cut-off low event (October 21-24).

[Figure]

**Figure S7: Averages of simulated reactive aromatics emitted from Tibet (red), India and nearby countries ("South Asia", blue) and China excluding Tibet ("China w/o Tibet", green) corresponding to in situ observations during October 19-20 and October 21-24. REAM simulations are conducted with original emissions (a) and the aromatics emissions redistributed following the BC emission distribution (b), respectively.**

**Thus, we conclude that the BC transport is enhanced by the cut-off low. We now clarify these issues in Section 3.3 as follows:**

"Compared to aromatics, BC is also subject to wet scavenging, which greatly reduces its transport efficiency by convection. During our analysis period, the cut-off low system and the associated precipitation are to the northwest of Tibet (Fig. S6 in the Supplement). Precipitation south of Tibet is weak and thus the subsequent removal of BC during trans-Himalaya transport is limited.

To examine the sensitivity of trans-Himalaya transport to the distribution of emission sources, we redistribute INTEX-B the total aromatics emissions over China and other South Asia countries on the basis of the MIX BC emission distributions. We conduct a sensitivity simulation using the redistributed emissions and compared the results to the original simulation. The trans-Himalaya transport from South Asia clearly dominates and it is strongly affected by the presence of a cut-off low system during our analysis period (Fig. S7 in the Supplement). Our analysis implies that BC transported in the presence of an upper tropospheric cut-off low is potentially a major contributor to BC deposition to Tibetan glaciers."

*2. There is a lack of recognition of previous studies, especially of regional chemistry transport modeling in South Asia and western China. Some previous papers to consider are listed in the references.*

**We did another search of related literature and discussed the relevant studies:**

 "The deposition of BC on the vast glaciers of the Tibetan Plateau will decrease the surface albedo, accompanied by increased sunlight absorption and subsequent enhanced melting (Hansen and Nazarenko, 2004; Ramanathan and Carmichael, 2008; Ming et al., 2009; Yasunari et al., 2010). Increasing BC concentrations were previously found in ice core and lake sediment records (Xu et al., 2009; Cong et al., 2013). The dwindling of glaciers over Tibet is a major concern for fresh water supply to a large portion of the Asian population through the Indus River, Ganges River, Yarlung Tsangpo River, Yangtze River and Yellow River (Singh and Bengtsson, 2004; Barnett et al., 2005; Lutz et al., 2014). Though melting glaciers favor river runoff temporarily, mass loss of glaciers endangers water supply during the dry season in the future (Yao et al., 2004; Kehrwald et al., 2008).

Increasing BC concentrations were already found in ice core and lake sediment records (Xu et al., 2009; Cong et al., 2013 Besides narrowing the uncertainties of BC emissions, aging and deposition, better understanding the transport pathways are equally important in this region. Surrounded by the largest BC sources of East Asia and South Asia (Bond et al., 2007; Ohara et al., 2007), Tibet is primarily affected by pollutant transport from these two regions (Kopacz et al., 2011; Lu et al., 2012; Zhao et al., 2013; Wang et al., 2015; Zhang et al., 2015; Li et al., 2016; Wang et al., 2016; Kang et al., 2016). Kopacz et al. (2011) attempted to identify the sources of BC over glaciers in the Himalayas and the Tibetan Plateau (HTP) using the adjoint model of GEOS-Chem. Lu et al. (2012) developed a novel back-trajectory model with BC emissions, hydrophilic-to-hydrophobic aging, and deposition and found that South Asia and East Asia account for 67% and 17% of BC over the HTP. Using source tagging, biofuel and biomass burning emissions from South Asia are found to be the largest sources of BC in HTP followed by fossil fuel combustion emissions (Zhang et al., 2015). Hindman and Upadhyay (2002) suggested that the vertical lifting due to convection and subsequent horizontal mountain-valley wind lead to the transport of aerosols from Nepal to Tibet. Dumka et al. (2010) also stressed the importance role of mountain-valley wind in BC concentration in Central Himalayas. Cong et al. (2015) suggested that both the large-scale westerlies from South Asia and the local mountain-valley wind from South Asia are major transport pathways. The synoptic scale trough and ridge can potentially lead to the trespassing of atmospheric brown clouds from South Asia to the Tibetan Plateau (Lüthi et al., 2015). Ji et al. (2015) indicated that the southwesterlies during monsoon season favor aerosols transport across the Himalayas from South Asia. Aerosols observations in previous studies are mostly limited to the southern and northern slopes of the Himalayas (Hindman and Upadhyay, 2002; Dumka et al., 2010; Cong et al., 2015) with very few in situ sites (e.g. Namco, Linzhi) inside Tibet (Kopacz et al., 2011; Ji et al., 2015; Lüthi et al., 2015; Zhang et al., 2015). Considering the complex topography (Lawrence and Lelieveld, 2010; Ménégoz et al., 2013; He et al., 2014; Kumar et al., 2015) and scarce observations (Maussion et al., 2011), it is crucial to evaluate model simulated transport performance over the Tibetan Plateau using available observations with a good spatial coverage. Observation-constrained modeling is needed to better understand potential model biases due to the uncertainties of model simulated transport from South Asia to Tibet."

*3. When figures are discussed in enough detail, it is better to place them in the main part of the paper. In my opinion, the supplement should not contain information that is needed to support the conclusions of the paper. For example, Figure S2 should be part of the main paper because it supports the conclusion that the INTEX-A aromatics emission estimates are much lower than values determined from a top-down estimate.*
*Please write the paper so that the reader can easily understand the main points of the study.*

**Thank you for the suggestion. We now combined Fig. S2 and Fig. 2. Please note that we did not use INTEX-A emission estimates in this work. Only INTEX-B data are used. Per reviewer's suggestion (see the response below), we updated emissions using the MIX inventory (please see updates in section 2.3).**

[Figure]

**Figure 2: SCIAMACHY observed  (a), REAM simulated (b) CHOCHO VCDs, the low bias of simulated CHOCHO VCDs (c), simulated isoprene (d) and aromatics (e) contributions to CHOCHO VCDs using the a priori emissions for October 2010. White areas  denote missing satellite data or ocean. For each valid SCIAMACHY data point, a corresponding model value is sampled in (b) and (c).**

*Minor Comments:*

*1. Page 4, Line 1: How does one choose $\alpha$ for use in the uncertainty equation? Is this even necessary information for the reader?*

**The range of $\alpha$ shows the relative uncertainty of SCIAMACHY glyoxal (CHOCHO) VCDs. This will help readers understand that the retrieval uncertainty cannot explain the discrepancy between observed and simulated CHOCHO VCDs. This stresses the needs for the top-down emission estimate.**

**We add the following sentence to Section 2.4.**

"First, we calculate the difference between observed ($C_{CHOCHO}^{SCIAMACHY}$, Fig. 2a) and modeled ($C_{CHOCHO}^{REAM}$, Fig. 2b) CHOCHO VCDs with original emissions ( $\Delta C_{CHOCHO} = C_{CHOCHO}^{SCIAMACHY} - C_{CHOCHO}^{REAM}$ , Fig. 2c). This discrepancy greatly exceeds the uncertainties of SCIAMACHY retrieval."

*2. I suggest rearranging the model description paragraphs. How would you describe the model to someone who has never worked with regional transport models? I suggest beginning with the CFSR dataset because it is used to provide initial and boundary conditions for the WRF model. Then the WRF model should be described, citing what version, resolution, and parameterizations are used. Next, it should be stated how REAM takes information from the WRF simulation. Does it take WRF output every hour, every 3 hours, etc.? Finally, the REAM model should be described. Do not rely on the reader to go to the cited references to get needed information, but instead to go to the cited references to get more details.*

**Thank you for the suggestion. We now add more details of the model.**

"REAM has a horizontal resolution of 36 km with 30 vertical levels in the troposphere and 5 vertical levels in the stratosphere covering adjacent regions of China (Fig. 1a1b). The model top is at 10 hpa. Meteorological fields in REAM are obtained from the Weather Research and Forecasting model (WRF) assimilations constrained by National Centers for Environmental Prediction Climate Forecast System Reanalysis (NCEP CFSR, Saha et al., 2010) 6-hourly products, which have a horizontal resolution of T382 (~38 km). We run the WRF model with the same resolution as in REAM with a domain is larger than that of REAM by 10 grid cells on each side. Meteorological inputs related to convective transport are updated every 5 minutes while the others are updated every 30 minutes. The recent update of REAM expands the GEOS-Chem standard chemical mechanism (V9-02) to include a detailed description of aromaticaromatics chemistry (Bey et al., 2001; Liu et al., 2010, 2012b). Aromatics are lumped into three species based on reactivity, i.e. ARO1 (toluene, ethyl-benzene), ARO2 (m/p/o-xylene), and benzene. The atmospheric lifetimes of the three aromatics tracers against OH are 18 hours, 4.2 hours and 3.9 days during the study period (October 13-25, 2010), respectively. Due to the long atmospheric lifetime of benzene, it is more difficult to track and identify its sources; thus we do not explicitly discuss benzene in this study. We focus our analysis on reactive aromatics (toluene, ethyl-benzene, and m/p/o-xylene).

Meteorological fields in REAM are obtained from the Weather Research and Forecasting model (WRF) assimilation constrained by National Centers for Environmental Prediction Climate Forecast System Reanalysis (NCEP CFSR, Saha et al., 2010). CFSR has a horizontal resolution of T382 (~38 km). Initial and boundary conditions for chemical tracers are taken from GEOS-Chem (V9-02) 2° × 2.5° simulation. (Bey et al., 2001)."

*3. It is important to include what the model top is because of the high surface elevation of Tibet that is prone to have stratospheric intrusions (perhaps falsely if the model top is too low).*

**The model top is 10 hpa, which is above the tropopause. REAM has 30 layers in the troposphere and 5 layers in the stratosphere. So it should be able to resolve stratospheric intrusions. We added this information to Section 2.3.**

"REAM has a horizontal resolution of 36 km with 30 vertical levels in the troposphere and 5 vertical levels in the stratosphere covering adjacent regions of China (Fig. 1a1b). The model top is at 10 hpa."

*4. Has the REAM model been evaluated for the region simulated? In this paper we see comparisons with SCIAMACHY and ground-based observations. How does the model perform in terms of meteorology and chemical constituents, such as CO, O3, NOx, and particulate matter?*

**Unfortunately, we do not have many in situ observations in the region (in particular Tibet) to evaluate the model. In the introduction section, we discussed previous studies and noted the lack of in situ observations over Tibet (very few surface sites). The dataset reported in this study provides valuable information over the Tibetan Plateau. In an effort to respond to reviewer's comment, we make a comparison with available satellite products. It appears that REAM simulated $NO_2$ and CO compared reasonably well with KNMI DOMINOv2 tropospheric $NO_2$ VCDs (http://www.temis.nl/airpollution/no2col/) and MOPITT daytime total CO VCDs (http://www.acom.ucar.edu/mopitt/MOPITT/) for Tibet, India and nearby regions (Fig. S2) during the period of the study. For the general evaluation of the model, we refer to other publications using REAM cited in Section 2.3.**

[Figure]

**Figure S2:** **MOPITT retrieved (a) and REAM simulated (b) monthly averaged total CO VCDs during October 2010. OMI retrieved (c) and REAM simulated (d) monthly averaged tropospheric NO₂ VCDs during October 2010. White areas denote missing data. MOPITT data are from http://www.acom.ucar.edu/mopitt/MOPITT/. OMI data are from http://www.temis.nl/airpollution/no2col/. Averaging kernels are applied to the model results.**

**We now discuss REAM performance in Section 2.3:**

"Compared with satellite observed CO and NO₂ VCDs, REAM performs reasonably well in the study region during October 2010 (Fig. S2 in the Supplement). For general model evaluations of REAM, we refer the readers to the papers cited early in this section.
We updated the INTEX-B emission inventory in South Asian countries through inverse modeling constrained by SCIAMACHY CHOCHO VCDs (next section)."

*5. The model domain is shown in Figure 1, allowing the reader to recognize that the region of interest is mostly away from the model boundaries. Is the Tibet region affected by the composition outside the model boundaries (especially the western boundary), or outside the subdomain shown in Figure 2?*

**Since the regions outside the model boundary are either remote or too far away from the studied regions, the contributions from these regions to both glyoxal and aromatics are negligible. Also, we use the concentrations from a global chemistry transport model (GEOS-Chem) as our boundary conditions.**

*6. When comparing the REAM model results with the SCIAMACHY satellite retrieval of glyoxal, is the model sampled the same way as SCIAMACHY sees the atmosphere?*
*For example, I assume that the missing data in Figure 2a from the satellite data is due to clouds. Are cloudy grid points removed from the model analysis (it doesn't appear so since there are no "missing data" from the model results).*

**We resampled REAM** in **the results in Fig. 2 (shown in major comment #2). We added in the caption of Fig. 2** "**For each valid SCIAMACHY data point, a corresponding model value is sampled in (b) and (c).**" **Interference**s **of clouds and water vapor have been removed in the SCIAMACHY retrieval. The missing data in satellite retrieval results from quality control in retrieval algorithm, which can result from excessive cloud coverage, large uncertainty and measurement failure. Glyoxal VCDs in Tibet are very low and the relative error is large. Consequently, these values are removed. Considering the retrieval uncertainties over the Tibetan Plateau, we only perform top-down emission inversion in regions south of the Himalayas.**

*7. Page 5, Line 21. It would be helpful to see a MEGAN emissions map of isoprene for the region.*

**We revise this sentence "The high isoprene contribution to glyoxal VCDs is to the southeast of the Indo-Gangetic Plain, where CHOCHO VCDs are high in both the observations and model simulations." Fig. 2d shows the contribution from isoprene to glyoxal VCDs.**

*8. Page 5, Lines 16-24. It may be helpful to include the glyoxal chemistry in the supplement.*

**We stated "CHOCHO is produced primarily from the photochemical oxidation of biogenic compounds (e.g., isoprene and terpenes) and hydrocarbon released by anthropogenic activities (e.g., acetylene, ethylene, and aromatics) (Fu et al., 2008)." The relevant chemistry is already discussed by Fu et al. (2008).**

*9. Section 2.4. Why is the INTEX-B emissions inventory, which is appropriate for year 2006, being used for the model simulation for year 2010? MACCity emissions (appropriate for 2010) or EDGAR-HTAP emissions may have been better suited for these simulations. Could the authors discuss the differences between the emissions inventory that they used and these more recent emissions inventories?*

**Thank you for the suggestion. We currently use MIX emissions inventory (Li et al., 2015) with INTEX-B aromatics emissions for countries other than China. HTAP-EDGAR now incorporates MIX for Asian emissions. MIX is mosaic emission inventory with MEIC for China and several other emission inventories for other countries. In addition to the MIX inventory, we also conduct sensitivity simulations using the Intercontinental Chemical Transport Experiment-Phase B (INTEX-B) emissions inventory (Zhang et al., 2009; Li et al., 2014), which was developed for the year 2006. We find that compared to the in situ observations of aromatics, the simulation results using the INTEX-B emissions are better. The main reason for the simulation improvements is due to the emissions of aromatics in South Asia. Given the large uncertainties in the emissions of aromatics (e.g., Liu et al., 2012b), this result is not surprising. Since MEIC and INTEX-B inventories are developed by the same group, we replace MIX aromatics emissions outside China with INTEX-B data such that aromatics emissions in the model are consistent. The improvements of model simulations compared to in situ observations are shown in Fig. S1. Since satellite observations are used to improve aromatics emissions (next section), using either MIX or INTEX-B emissions in this work gives the same conclusions.**

**MACCity emissions inventory is a linear combination of Atmospheric Chemistry and Climate Model Intercomparison Project (ACCMIP) and Representative Concentration Pathway (RCP) 8.5. MACCity is essentially RCP 8.5 for year 2010 and is not suitable for this research.**

[Figure]

**Figure S1: Comparisons between REAM simulated and in-situ observed reactive aromatics concentrations with (a) and without (b) INTEX-B aromatics emissions for countries excluding China.**

**We now discuss the use of emission inventories in Section 2.3 as follows.**

"Initial and boundary conditions for chemical tracers are taken from GEOS-Chem (V9-02) 2° × 2.5° simulation. (Bey et al., 2001). Anthropogenic emissions in China and other Asian countries are from the are from the MIX inventory for October 2010 (Li et al., 2015). MIX is a mosaic Asian anthropogenic emission inventory with the Multi-resolution Emission Inventory for China (MEIC v1.0) for 2010 and updated ) for China and several other emission inventories for other Asian countries. In addition to the MIX inventory, we also conduct sensitivity simulations using the Intercontinental Chemical Transport Experiment-Phase B (INTEX-B) emission for 2006 (emissions inventory (Zhang et al., 2009; Li et al., 2014), respectively. which was developed for the year 2006. We find that compared to the in situ observations of aromatics, the simulation results using the INTEX-B emissions are better. The main reason for the simulation improvements is due to the emissions of aromatics in South Asia. Given the large uncertainties in the emissions of aromatics (e.g., Liu et al., 2012b), this result is not surprising. Since MEIC and INTEX-B inventories are developed by the same group, we replace MIX aromatics emissions outside China with INTEX-B data such that aromatics emissions in the model are consistent. The improvements of model simulations compared to in situ observations are shown in Fig. S1 in the Supplement. Since satellite observations are used to improve aromatics emissions (next section), using either MIX or INTEX-B emissions in this work gives the same conclusions."

*10. Section 3.1. It would be interesting to learn in more detail what the surface elevation is at the observation points and at the matching REAM model grid cells. Could there be discrepancies between model and observations because the model does not adequately represent the surface elevation?*

**The surface elevations between WRF results and observation during Periods 1 and 2 are similar. This is not surprising considering the generally smooth topography in these regions. We expect the winds to vary little within a WRF grid in such regions. As Fig. 7 shows, WRF simulated winds match in-situ**

observations during Period 2. We suggest that the underestimation of WRF winds during Period 1 results from a missing cut-off low system. We also discuss the potential relationship between observed reactive aromatics and standard deviations of surface elevations during Period 3 in Section 3.4, the latter region has more variable topography than Periods 1 and 2. We suggest that the underestimation of the model results from model incapability to resolve complex terrains.

*11. Page 7, Line 6. How are the source attributions computed?*

**For the sensitivity simulations, we only turn on aromatics emissions in one of the three regions, i.e. Tibet, other provinces of China and India and nearby regions. All the sensitivity simulations utilize OH concentrations specified to the archived values of the full model simulation using top-down emissions. The results from these simulations are used to calculate the source attributions.**

**We now clarify this in Section 2.3.**

"We further carried out three model sensitivity tests to calculate the contributions to surface aromatics from emissions over Tibet, other provinces of China, and South Asia (India and nearby regions). Each simulation is run with only the aromatics emissions from the corresponding region. The OH concentrations in each simulation are specified to the archived values of the full model simulation. The results for two sub-periods of Period 2 are examined in Section 3.3."

*12. Page 7, Lines 15-17. Could the "cutoff low system" be described in more detail? Would "closed low" be a more appropriate term? (see the NWS definition at http://forecast.weather.gov/glossary.php?word=cutoff%20low) How long did the cutoff low remain in the region? Was there precipitation associated with the cutoff low?*

**Thanks for the reference. This low system became fully detached from the westerlies after generation. The cut-off system stayed for about 3 days and dissipated. The cut-off system and associated precipitation are to the northwest of Tibet (Fig. S6b in the Supplement). We add clarification in Section 3.3.**

"During October 21-24, the presence of a southeastward-moving upper tropospheric cut-off low system induces increasingly stronger surface wind from India to Tibet (Fig. 5b, Hoskins et al., 1985). The cut-off low system is a closed low-pressure system detached from the westerlies. It began to form on October 21 and started to dissipate on October 24."

*13. Page 7-8. It would be helpful to see Figure S5 showing both Period 1 and Period 2. From what is presented, it is unclear whether WRF simulates the cutoff low pressure system (unless these are WRF results, which is not clear from the figure caption).*

**Fig. 5 shows the results for Period 2. In Fig. S8 (previously Fig. S5), both CFSR and CFSR-constrained WRF show a trough rather than a cut-off low system (Fig. 5) in Central Asia during Period 1.**

*14. Page 8, End of section 3. There should be a section added, discussing the results found in this study with previous papers (such as those listed in the references). For example, the Kumar et al. (2015) study also mentions the challenges of modeling pollutant transport in the Himalayas. Ji et al. (2015) also discuss aerosol transport from the IGP to Tibet.*

**We now discuss the related previous researches in Sections 3.3 and 3.4, respectively.**

**Section 3.3:**

"Accompanying this transport, large amounts of pollutants such as reactive aromatics analyzed here are transported to the Tibetan Plateau leading to much higher surface concentrations.

The cut-off low system provides a more rapid and efficient pollutants transport pathway compared with transport pathways previously proposed by other studies, such as westerlies (Cong et al., 2015; Ji et al., 2015) and mountain-valley winds (Hindman and Upadhyay 2002; Dumka et al., 2010)."

**Section 3.4:**

"Model resolution as high as 1 km appears to be necessary to capture the observed feature but the computational resource requirement will be exceptionally large for a global model such as that used for CFSR. Other issues related to complex terrains in this region were also discussed by previous studies (Maussion et al., 2011; Ménégoz et al., 2013; He et al., 2014; Kumar et al., 2015)."

*15. Figure 5. What is the source of the information plotted in Figure 5? Is it from the model (WRF + REAM) simulation? Please clarify. Why are the surface winds and simulated reactive aromatics shown only for Tibet?*

**Yes, the winds and geopotential heights come from WRF simulation while simulated reactive aromatics are REAM simulation results. We clarified this in the revised paper.**

**Fig. AR1 (below) shows surface winds and simulated reactive aromatics for the whole region. The surface winds can be misleading due to the high altitude of the Tibetan Plateau compared with surrounding regions.**

**To help readers focus on the study region and avoid confusion, we only show surface winds and simulated reactive aromatics over Tibet as in Fig. 5 in the paper.**

[Figure]

**Figure AR1: Distributions of WRF simulated surface wind and REAM simulated concentrations of reactive aromatics over the Tibetan Plateau during October 19-20, 2010 (a) and October 21-24, 2010 (b). Circles show the observed reactive aromatics concentrations. Composite distributions of simulated reactive aromatics concentrations and surface wind, corresponding to sampling time of the observations, are shown in color and by arrows, respectively. Corresponding WRF simulated 300 hPa**

**geopotential height fields are shown by contour lines. The border of Tibet Autonomous Region is colored green.**

[Figure]

Figure 5: Distributions of **WRF simulated** surface wind and **REAM simulated** concentrations of reactive aromatics over the Tibetan Plateau during October 19-20, 2010 (a) and October 21-24, 2010 (b). Circles show the observed reactive aromatics concentrations. Composite distributions of simulated reactive aromatics concentrations and surface wind over Tibet, corresponding to sampling time of the observations, are shown in color and by arrows, respectively. Corresponding **WRF simulated** 300 hPa geopotential height fields are shown by contour lines. The border of Tibet Autonomous Region is colored green.

*Technical Comments:*
*1. Page 1, Line 16: verb agreement: Long-range transport followed by deposition of black carbon on glaciers of Tibet is one of the key issues*
**Revised as suggested.**

*2. Page 1, Line 17: impacting the melting of glaciers*
**Corrected.**

*3. Page 1, Line 28: Remove "Furthermore"*
**Revised as suggested.**

*4. Page 2, Line 2: The Menon et al. (2002) paper should be cited.*
**Reference added.**

*5. Page 2, Line 5: Insert "altitude" or "surface elevation" after "4 km"*
**Corrected.**

*6. Page 2, Line 14 is an orphan sentence and is not really needed.*
**We moved this sentence.**

*7. Page 2, Line 19: "aerosols" may be a better word than "condensation nuclei"*
**Corrected.**

*8. Page 2, Line 20: "large-scale westerlies from East Asia" does not make sense. East Asia is east of Tibet, so it must be an easterly wind if the air moves east to west.*
**Corrected, it should be South Asia instead of East Asia.**

*9. Page 3, Line 20: Shouldn't Fig. 1a be cited before Fig. 1b?*
**Corrected.**

*10. Page 3, Line 23: I think it should be "overpass time" and not "overpassing".*
**Corrected.**

*11. Page 4, Lines 4-8: Are all these references needed? It is sufficient to just cite 1-2 example references per topic.*
**These previous applications are relevant since they provide observation-based evaluations and applications to the REAM model. Successful applications in various environments demonstrate the capability of the model.**

*12. Page 4, Line 24: It would be good to cite Figure S1a.*
**We cite Fig. S3a (previously Fig. S1) in Section 2.4.**

*13. Page 5, Lines 2-4 is a long sentence. Please break it up into 2 sentences.*
**Revised as suggested.**

*14. Page 5, Line 2: I think it should be "overpass time" and not "overpassing".*
**Revised as suggested.**

*15. Page 5, Line 28: I think it should be "overpass time" and not "overpassing".*
**Revised as suggested.**

*16. Page 5, Line 29: Insert "did" before "for eastern China".*
**Corrected.**

*17. Page 7, Line 15: I would suggest using "promote" instead of "provide".*
**Revised as suggested.**

*18. Page 7, Line 24: Are the histograms for wind speed for at the surface (or 10-m winds)? Please clarify.*
**The histograms are for surface winds. We now clarify in the paper.**

"Fig. 7 shows the histograms of observed and simulated surface wind speed for the 3 periods."

*19. References: Could the references be written so that they are easier to read? Either adding a "hanging indent" or a line space between references would help immensely.*
**Corrected, we now added a "hanging intent" for each reference.**

*20. Figure 4b: The black and dark blue colors are quite similar. Could a different color*

*be plotted?*
**We now used red and blue colors as suggested.**

*21. Figure S3: To emphasize the differences between the panels, it may be better to plot using the same scaling. The gradients can still be appreciated if a "log type" scaling is used, e.g. 1, 2, 3, 5, 7, 10.*
**We revised Fig. S4 (previously Fig. S3) as suggested.**

*22. Figure S7: The legend mistypes "original". The original winds line does not look like the black line in Figure S4d.*
**The legend has been corrected. Black line in Fig. S10 (previously Fig. S7) shows ground-1km air mass fluxes during Period 1. Fig. S5d (previously Fig. S4d) shows ground-1km air mass fluxes during October 19-20 and October 21-24.**

**Response to Referee #2:**

*First of all, the Authors claim that their analysis has implications for improving the modelling of black carbon (BC) transport to the glaciated regions of Tibet (all the Introduction is dedicated to this topic). However, their approach is based on measurements (in situ and satellite retrievals) of aromatic HCs and of their degradation products (glyoxal). It is certainly true that aromatic hydrocarbons share with BC several emission and transport patterns, but only to a certain extent. For instance, the aromatic HCs are emitted by fossil fuel combustion, gasoline evaporation and solvent use (page 2, line 24), however only the first of these three sectors is of importance for BC. It follows that top-down methods for correcting the emissions of aromatic HCs (Section 2.4) has unclear implications for improving the representation of BC sources in the models. If the scope of the paper is really improving BC modelling in the Himalayan-Tibetan region, then the absence of BC observations poses a major caveat, even if the approach is conceptually valid and in principle it could be extended to experiments involving real BC measurements.*

**The implications of this paper for BC are only on transport. As the reviewer pointed out that our work cannot directly address the accuracy of BC emission inventories, which we did not claim in the paper that we can either. We suggest that aromatics observations are good proxies for understanding transport processes of BC to the Tibetan Plateau. The major pathway of transport is driven by the presence of a cut-off low system. Section 2.4 shows that the underestimates of aromatics transport are due to an underestimation of emissions, which can be improved using satellite observations.**

**To further demonstrate the link between transport of aromatics and BC to Tibet, we redistributed the total aromatics emissions over China and other South Asia countries on the basis the BC emission distributions. Therefore, the resulting aromatics emission distributions resembles that of BC. We conducted a sensitivity simulation using these emissions and compared the results to the original REAM simulation (Fig. S7). Transport of BC from South Asia (e.g., India) clearly dominates and it is strongly affected by the presence of a cut-off low system, as we discussed in the paper.**

[Figure]

**Figure S7: Averages of simulated reactive aromatics emitted from Tibet (red), India and nearby countries ("South Asia", blue) and China excluding Tibet ("China w/o Tibet", green) corresponding to in situ observations during October 19-20 and October 21-24. REAM simulations are conducted with original emissions (a) and the aromatics emissions redistributed following the BC emission distribution (b), respectively.**

**We now discuss this as well as the effects of BC wet deposition in Section 3.3.**

"Compared to aromatics, BC is also subject to wet scavenging, which greatly reduces its transport efficiency by convection. During our analysis period, the cut-off low system and the associated precipitation are to the northwest of Tibet (Fig. S6 in the Supplement). Precipitation south of Tibet is weak and thus the subsequent removal of BC during trans-Himalaya transport is limited.
To examine the sensitivity of trans-Himalaya transport to the distribution of emission sources, we redistribute INTEX-B the total aromatics emissions over China and other South Asia countries on the basis of the MIX BC emission distributions. We conduct a sensitivity simulation using the redistributed emissions and compared the results to the original simulation. The trans-Himalaya transport from South Asia clearly dominates and it is strongly affected by the presence of a cut-off low system during our analysis period (Fig. S7 in the Supplement). Our analysis implies that BC transported in the presence of an upper tropospheric cut-off low is potentially a major contributor to BC deposition to Tibetan glaciers."

*Specific comments: a. Biomass burning is ruled out from the possible explanations for the difference between observed and retrieved glyoxal concentrations over the IGP, because satellite fire counts show only spot fire occurrence over the Plain with little correspondence with the model-measurement gap (Page 5, lines 10 – 14). However, open burning accounts for only a fraction of biomass burning, which is normally practiced also indoor for cooking, heating etc., undetected by remote sensing. Therefore, I would not rule out the hypothesis of a direct emission of glyoxal from domestic biomass burning.*

**We calculate indoor burning glyoxal emissions using emission factors from Pettersson et al. (2011) and Li et al. (2014). The Indian rural and urban population distribution from the NASA Socioeconomic Data and Applications Center (SEDAC) Network for year 2010 is used as spatial proxies. We adopt the energy consumptions for rural and urban inhabitants on the basis of the National Sample Survey Office of India (N.S.S.O., 2012a, 2012b). Simulated indoor burning contribution (Fig. S3) to glyoxal VCDs is lower by a factor of about 15 than the glyoxal VCDs discrepancy between satellite retrieval and model simulation. The uncertainty of the indoor energy consumption glyoxal emissions mainly results from the uncertainty of the emission factor. Even we assume this uncertainty to be 300%, indoor burning cannot explain the low bias of the simulated glyoxal VCDs. Thus, we consider that the underestimation of aromatics emissions is the main reason for the glyoxal VCD discrepancy. After accounting for model transport biases, the comparison of the model simulations using a priori emissions to in situ observations of reactive aromatics suggests that aromatics emissions in South Asia are underestimated.**

[Figure]

(a) Outdoor biomass burning contribution

(b) Indoor burning contribution

CHOCHO VCDs ($\times 10^{13}$ molec cm$^{-2}$)

**Figure S3: Contributions to CHOCHO VCDs from outdoor biomass burning (a) and indoor burning (b) emissions for October 2010.**

**We now discuss this in Section 2.3 and Section 2.4.**

**Section 2.3:**

"Biogenic VOC emissions are computed with the Model of Emissions of Gases and Aerosols from Nature (MEGAN) algorithm (v2.1, Guenther et al., 2012) and outdoor biomass burning emissions of CHOCHO and other species are based on Global Fire Emissions Database Version 4.1 with small fires (GFED4.1s, van der Werf et al., 2010; Andreae and Merlet, 2001; Lerot et al., 2010). Indoor burning CHOCHO emissions of India are computed using emission factors from Pettersson et al. (2011) and Li et al. (2014). Rural and urban population distributions of India for year 2010 are used as spatial proxies (Balk et al., 2006; CIESIN, 2011, 2016). We adopt the energy consumptions for rural and urban inhabitants on the basis of the National Sample Survey Office of India (N.S.S.O., 2012a, 2012b)."

**Section 2.4:**

"The contribution to CHOCHO VCDs from outdoor biomass burning (Fig. S3a in the Supplement) differs greatly from that of $\Delta C_{CHOCHO}$ (Fig. 2c), which is large over the industrialized Indo-Gangetic Plain. Simulated indoor burning contribution to CHOCHO VCDs is lower by a factor of about 15 than the CHOCHO VCDs discrepancy between satellite retrieval and model simulation (Fig. S3b in the Supplement). The uncertainty of the indoor burning CHOCHO emissions mainly results from that of the emission factor. Even we assume this uncertainty to be 300%, indoor burning cannot explain the low bias of the simulated CHOCHO VCDs. Therefore,  the large model underestimation of CHOCHO over the Indo-Gangetic Plain is unlikely due to outdoor biomass burning or indoor burning during our analysis period."

*b. The Authors find a plausible explanation for the rise of aromatic HCs concentrations between 22 and 24 Oct 2010 in the synoptic meteorological conditions in central Asia showing an upper-level cut-off system triggering a southerly circulation from India to Tibet. However, minding that BC can be removed during transport by precipitations, the Authors should provide a more in-depth analysis of the meteorological conditions over the Himalayans during the approach of the low-pressure system. Apparently, on the 22 of*

*October, frontal cloud systems travelled over the Tibet from west to east (http://www.ssec.wisc.edu/data/comp/ir/2010295M0000.gif). The presence of precipitations with possible losses of BC (and not necessarily of aromatic HCs) in the Himalayas should be checked carefully at local meteorological stations.*

**This is a good point for BC concentration distribution over the Tibet, which is not studied in this work. Our focus is on transport. BC still needs to be transported to the Tibetan region; scavenging can only reduce the BC amount. If BC is not transported to the region in the first place, scavenging cannot increase BC in this region.**

**Our study suggests that the cut-off low system is difficult to simulate with a regional model constrained by meteorological observations. It would be much harder for climate models to simulate correctly. Fig. S6 in the Supplement shows the precipitation distribution for October 19-20 and October 21-24. The cut-off system is to the northwest of Tibet. Precipitation south of the Himalayas is weak and hence the removed BC from precipitation during the trans-Himalaya transport is limited.**

[Figure]

**Figure S6: WRF simulated averaged daily precipitation for October 19-20 (a) and October 21-24 (b), respectively.**

**We now discuss the related issues in Section 3.3.**

[revised manuscript text omitted]

---

## Author Response (AR2)

We appreciate the editor and the two reviewers for their insightful comments and suggestions concerning our research. We made technical corrections accordingly. We present our response and changes below. The editor's and the reviewers' comments and suggestions are shown in *italic*. Authors' responses are in **bold**.

**Response to Editor:**

*I consider that the revision has addressed the issues raised by the reviewers.*
*Both of them agree on publication indicating minor changes.*
*I would require that Authors address them in the final version of the manuscript. In particular, it would be important to add guidance to the reader on the general applicability of the results related to multiple processes affecting BC distribution, focusing on wet deposition / precipitation effects.*

*- Add a sentence on BC precipitation scavenging while discussing hygroscopic growth as outlined by rev. 1*

**We now clarify this issue in Section 3.3.**

"The cut-off low system provides a more rapid and efficient pollutants transport pathway compared with transport pathways previously proposed by other studies, such as westerlies (Cong et al., 2015; Ji et al., 2015) and mountain-valley winds (Hindman and Upadhyay 2002; Dumka et al., 2010). Compared to aromatics, BC is also subject to wet scavenging, which greatly reduces its transport efficiency by convection. In-cloud BC scavenging is due to cloud activation or ice nucleation and subsequent removal by precipitation, and below-cloud scavenging is due to collision with rain droplets (e.g., Taylor et al., 2014). During our analysis period, the cut-off low system and the associated precipitation are to the northwest of Tibet (Fig. S6 in the Supplement). Precipitation south of Tibet is weak and thus the subsequent removal of BC during trans-Himalaya transport is limited."

*- Statement in the paper recommending to take into account all aspects of BC transport to the Tibetan plateau (including the distribution of wet depositions) when expanding the future analysis from a case-study basis to a climatological basis as suggested by rev. 2*

**We now add a statement as suggested in Section 4.**

"Our results imply that pollution transport to the Tibetan Plateau, such as that of BC, is likely to be greatly underestimated in climate models, which was found previously (e.g., He et al., 2014). In addition to trans-Himalaya transport, BC emissions, chemical transformation, and wet deposition also require extensive evaluations with the observations over the region. Further analysis of reanalysis and climate model simulations is required to quantify potential model biases and the resulting effect of simulated BC deposition to glaciers on the Tibetan Plateau due to the transport issues we identified in this study."

*Finally, include the suggestion of rev. 2 on sec. 3.3 wording.*

**Corrected.**

"To examine the sensitivity of trans-Himalaya transport to the distribution of emission sources, we redistribute the INTEX-B the total aromatics emissions over China and other South Asia countries on the basis of the MIX BC emission distributions."

**Response to Referee #1:**

*The authors have satisfactorily addressed my comments on their paper and have included needed changes to the manuscript and supplement.*

*While it is not necessary to include for this paper, a possible source of in situ ozone data:*
*Bian et al. (2012) http://onlinelibrary.wiley.com/doi/10.1029/2012GL052996/full*

**Thanks for the reference. The measurements were taken in August 2010 during the monsoon season. Hereby, we are not including these measurements in this study.**

*In regards to black carbon (BC) scavenging by precipitation, this process is not very well understood because there are not many good measurements quantifying BC scavenging. While I agree that BC hygroscopicity is important for scavenging, BC will also be subject to impaction scavenging by precipitation. Thus, when discussing BC precipitation scavenging, several processes, cloud drop activation, ice nucleation, and impaction of aerosols by precipitation, should be considered.*

**We now clarify this issue in Section 3.3.**

"The cut-off low system provides a more rapid and efficient pollutants transport pathway compared with transport pathways previously proposed by other studies, such as westerlies (Cong et al., 2015; Ji et al., 2015) and mountain-valley winds (Hindman and Upadhyay 2002; Dumka et al., 2010). Compared to aromatics, BC is also subject to wet scavenging, which greatly reduces its transport efficiency by convection. In-cloud BC scavenging is due to cloud activation or ice nucleation and subsequent removal by precipitation, and below-cloud scavenging is due to collision with rain droplets (e.g., Taylor et al., 2014). During our analysis period, the cut-off low system and the associated precipitation are to the northwest of Tibet (Fig. S6 in the Supplement). Precipitation south of Tibet is weak and thus the subsequent removal of BC during trans-Himalaya transport is limited."

**Response to Referee #2:**

*The manuscript has been greatly improved since the first submission. The new Fig S6 reporting the WRF-estimated precipitation fields supports the Author's hypothesis about BC being advected to the Tibet during cut-off low events without significant depositions on the southern Himalayan side. However, the analysis is based on a case study, and the removal of aerosols during the uplift may occur to a different extent during other events. I would like to read an Author's statement in the paper recommending to take into account all aspects of BC transport to the Tibetan plateau (including the distribution of wet depositions) when expanding the future analysis from a case-study basis to a climatological basis.*

**Thanks for the suggestion. We add a statement accordingly in Section 4.**

"Our results imply that pollution transport to the Tibetan Plateau, such as that of BC, is likely to be greatly underestimated in climate models, which was found previously (e.g., He et al., 2014). In addition to trans-Himalaya transport, BC emissions, chemical transformation, and wet deposition also require extensive evaluations with the observations over the region. Further analysis of reanalysis and climate model simulations is required to quantify potential model biases and the resulting effect of simulated BC deposition to glaciers on the Tibetan Plateau due to the transport issues we identified in this study."

*Technical correction on the new text in Sect. 3.3: "To examine the sensitivity of trans-Himalaya transport to the distribution of emission sources, we redistribute INTEX-B the total aromatics emissions over China and other South Asia countries on the basis of the MIX BC emission distributions." There is something wrong in the syntax of this phrase ("we redistribute INTEX-B the...").*

**Corrected.**

[revised manuscript text omitted]